# Machine learning reveals genes impacting oxidative stress resistance across yeasts

Katarina Aranguiz[1,2,11], Linda C. Horianopoulos [1,2,11], Logan Elkin [1,2,10], Kenia Segura Abá [3,4], Drew Jordahl [5,6], Katherine A. Overmyer[1,6], Russell L. Wrobel [1,2], Joshua J. Coon [1,5,6], Shin-Han Shiu [3,4,7,8], Antonis Rokas [9] & Chris Todd Hittinger [1,2,5] ✉

Reactive oxygen species (ROS) are highly reactive molecules encountered by yeasts during routine metabolism and during interactions with other organisms, including host infection. Here, we characterize the variation in resistance to the ROS-inducing compound *tert*-butyl hydroperoxide across the ancient yeast subphylum Saccharomycotina and use machine learning (ML) to identify gene families whose sizes are predictive of ROS resistance. The most predictive features are enriched in gene families related to cell wall organization and include two reductase gene families. We estimate the quantitative contributions of features to each species' classification to guide experimental validation and show that overexpression of the old yellow enzyme (OYE) reductase increases ROS resistance in *Kluyveromyces lactis*, while *Saccharomyces cerevisiae* mutants lacking multiple mannosyltransferase-encoding genes are hypersensitive to ROS. Altogether, this work provides a framework for how ML can uncover genetic mechanisms underlying trait variation across diverse species and inform trait manipulation for clinical and biotechnological applications.

Yeasts of the subphylum Saccharomycotina display profound diversity at both the genetic and phenotypic levels[1–3]. Their genetic diversity is comparable to that of the kingdom Animalia, allowing them to occupy every marine and terrestrial biome on Earth[1,4,5]. Despite this diversity, most yeast research has focused on the model yeast *Saccharomyces cerevisiae*, a selection of industrially relevant yeasts that include *Yarrowia lipolytica* and *Komagataella phaffii*, and pathogenic *Candida* species with an emphasis on the mechanisms by which they cause disease. Recent efforts highlight the increasing importance of considering trait variation among non-conventional yeasts for their potential contributions to biotechnology and a transition to a circular economy and bioeconomy[6,7], as well as for the early identification of emerging pathogens[8–10]. With the growing availability of yeast genomes[1,3,11–14], comparative genomics can be combined with high-throughput data collection to explore the genotype-phenotype map for traits of interest. Comparative approaches allow for the identification of the molecular bases of evolutionary events, such as mutations, gene losses or gains, and gene family expansions leading to the modification or acquisition of traits[9,15–18].

A trait of both clinical and biotechnological importance in yeasts is their resistance to reactive oxygen species (ROS). ROS are unstable, oxygen-containing molecules, such as $H_2O_2$ or $O_2^-$, that occur naturally

[1]DOE Great Lakes Bioenergy Research Center, University of Wisconsin-Madison, Madison, WI, USA. [2]Wisconsin Energy Institute, Center for Genomic Science Innovation, J. F. Crow Institute for the Study of Evolution, Laboratory of Genetics, University of Wisconsin-Madison, Madison, WI, USA. [3]DOE Great Lakes Bioenergy Research Center, Michigan State University, East Lansing, MI, USA. [4]Genetics and Genome Science Program, Michigan State University, East Lansing, MI, USA. [5]Cellular and Molecular Biology Graduate Program, University of Wisconsin-Madison, Madison, WI, USA. [6]Department of Biomolecular Chemistry, University of Wisconsin-Madison, Madison, WI, USA. [7]Department of Plant Biology, Michigan State University, East Lansing, MI, USA. [8]Department of Computational Mathematics, Science, and Engineering, Michigan State University, East Lansing, MI, USA. [9]Department of Biological Sciences and Evolutionary Studies Initiative, Vanderbilt University, Nashville, TN, USA. [10]Present address: Cell Biology, Neurobiology and Anatomy, Medical College of Wisconsin, Milwaukee, WI, USA. [11]These authors contributed equally: Katarina Aranguiz, Linda C. Horianopoulos. ✉e-mail: cthittinger@wisc.edu

in living organisms as a byproduct of electron transport chain activity, but they can be harmful in large amounts[19,20]. During industrial overproduction of lipids or proteins, which are produced in a manner that generates oxidative stress, ROS can limit bioproduct yields[21–23]. Exogenous ROS can also be encountered by yeasts through interactions with plants or animals as these organisms use ROS bursts to restrict microbial growth and limit infections[20]. The ability of pathogenic yeasts to withstand ROS influences the outcomes of their interactions with hosts, particularly for yeasts that must survive phagocytosis and ROS exposure in the phagosome[24,25]. Ultimately, we expect yeasts to frequently encounter ROS through interactions with other organisms, their natural environment, and human immune cells, which may collectively lead to the evolution of resistance mechanisms varying in strength and mode[20].

Efforts to describe genetic mechanisms underlying ROS resistance in yeast have identified key enzymes and regulators by screening genetic knockouts or measuring transcriptional responses in *S. cerevisiae*[26–28]. Similar efforts have been used to characterize the oxidative stress responses in other yeasts, such as *Kluyveromyces lactis*[29] and pathogenic *Candida* species[30,31]. Comparative studies focused on the genetic basis of ROS resistance across different yeast species are limited. A previous study compared stress response genes with *S. cerevisiae* orthologs across 18 fungal species and found that key components were highly conserved at the sequence level, but upstream sensors were more divergent[32]. There are also examples of comparisons of transcriptional responses to oxidative stress between *S. cerevisiae* and a few well studied species: *Schizosaccharomyces pombe, Candida albicans*[33], and *Lachancea kluyveri*[34]. Additionally, there have been several comparative studies describing phenotypic differences between yeast species during oxidative stress exposure, but these did not interrogate underlying genetic mechanisms[35–39]. Outside of *S. cerevisiae* and a handful of emerging model species, deep knowledge about ROS resistance in yeasts is lacking. Therefore, we hypothesized that using an experimental and comparative genomic approach would identify both new and previously underappreciated genetic mechanisms that contribute to ROS resistance broadly among yeasts or within specific species.

Describing genotype-phenotype connections is inherently challenging because there is rarely an exclusive and causal relationship between a single gene and a given phenotype. In genomics, machine learning (ML) can be applied as a tool that learns patterns among genetic features to make phenotypic predictions and identify genes underlying variation[40]. ML algorithms, such as Random Forest (RF), combined with feature selection techniques have proven to be effective tools for handling large datasets to describe genetic signatures of a given phenotype[41]. While the application of ML across strains or mutants of a single species is being employed widely[42–45], there are still relatively few studies that have used ML in a macroevolutionary context (i.e. assessing large-scale evolutionary patterns across millions of years, such as an entire subphylum), and these studies often use a specific set of enzymes or curated annotations as features[3,46–49].

In this study, we took a genome-wide comparative approach by training an ML model to identify gene families predictive of high resistance to the ROS-inducing compound *tert*-butyl hydroperoxide (TBOOH) across the yeast subphylum using all orthologous gene counts as features. We found that gene families encoding reductases were among our most predictive features, and the top 50 features were enriched for cell wall-related gene families. To estimate each feature's impact on each species' prediction, we used SHapley Additive exPlanations (SHAP) values; positive values indicate features that push a species toward a positive ROS-resistant classification, while negative values indicate the opposite[50]. SHAP values allowed us to prioritize gene-species combinations for functional validation, which is crucial to the application of ML findings. In validating these predictions, we confirmed that overexpression of the gene encoding old yellow

enzyme (OYE) increased ROS resistance in a susceptible yeast, *K. lactis*, and that deletion of two genes encoding mannosyltransferases decreased ROS resistance in the model yeast *S. cerevisiae*. Overall, our study generated a classification model that provided novel and functionally validated insights into the mechanisms of ROS resistance evolution through gene family expansions and highlighted groups of genes that are associated with ROS resistance across the yeast subphylum.

## Results

### Identification of ROS-resistant and ROS-sensitive species

To explore the phylogenetic distribution of ROS resistance levels, we analyzed the relative growth of 285 yeast species in two different concentrations of the ROS-generator TBOOH. ROS resistance varied widely across the yeast subphylum (Fig. 1, Supplementary Data 1) and was largely independent of taxonomic order[51] (formerly major clade[1]) when considering orders with at least ten species (Supplementary Fig. 1). In fact, it was common for closely related and even sister taxa to have drastically differing ROS resistance levels. Despite this, there were a few notable genera that displayed primarily resistant or sensitive phenotypes. Species within the genera *Yarrowia, Kazachstania*, and *Tetrapisispora* were highly ROS-resistant, and species within the genera *Metschnikowia, Lachancea*, and *Kluyveromyces* were generally highly ROS-sensitive. ROS susceptibility also varied among known pathogens, despite the importance of ROS resistance in surviving host defense mechanisms (Fig. 1). For example, *C. albicans* was ROS-resistant, whereas *C. auris* was ROS-sensitive. Although these data are intriguing, we note that these pathogens may respond to ROS differently in vivo than in our in vitro condition.

### A machine learning model predicts ROS resistance

Since these data revealed that the most resistant yeast species were widely dispersed throughout the yeast subphylum, we established an ML classification model to predict ROS resistance across yeast species using gene family size (based on the number of genes in each orthogroup) as features. To establish a biologically relevant classification model, we classified the poorest-growing 20% of species in 1 mM TBOOH as ROS-sensitive and the best-growing 20% of species in 2 mM TBOOH as ROS-resistant for a total of 114 species (Fig. 1B, C). The classification column was combined with the gene family size for each species (from the orthogroup matrix previously generated for the Saccharomycotina subphylum[3]). An RF feature selection algorithm was used to identify the 50 most important features based on Gini importance. Using these 50 features, an RF classifier was trained with 90% of species, and 10% were held out for testing. We found that the model successfully classified yeasts as evidenced by the area under the curve of the receiver operating characteristic (AUC-ROC) above 0.90 and F1 scores above 0.85 in both the validation and testing sets (Fig. 2). This result indicates that gene family size is predictive of ROS resistance.

### Functional characteristics of predictive gene families

The top two features of importance as ranked by the ML model were both gene families encoding reductases, specifically aldehyde reductases and OYE oxidoreductases (Fig. 3, Supplementary Data 2). Reductases catalyze the transfer of electrons to convert unstable molecules into less reactive ones, mitigating damage induced by ROS[52–55]. Although these genes have functions that can be easily related to ROS mitigation, we noted that there was not a sharp decrease in importance after these two features but rather a gradual and subtle decrease (Fig. 3A). Therefore, many of the other top features are also contributing to the correct classifications, suggesting that ROS resistance is complex and polygenic. This result prompted us to further interrogate the potential functions of the top 50 gene families by performing a gene ontology (GO) term enrichment analysis using all of

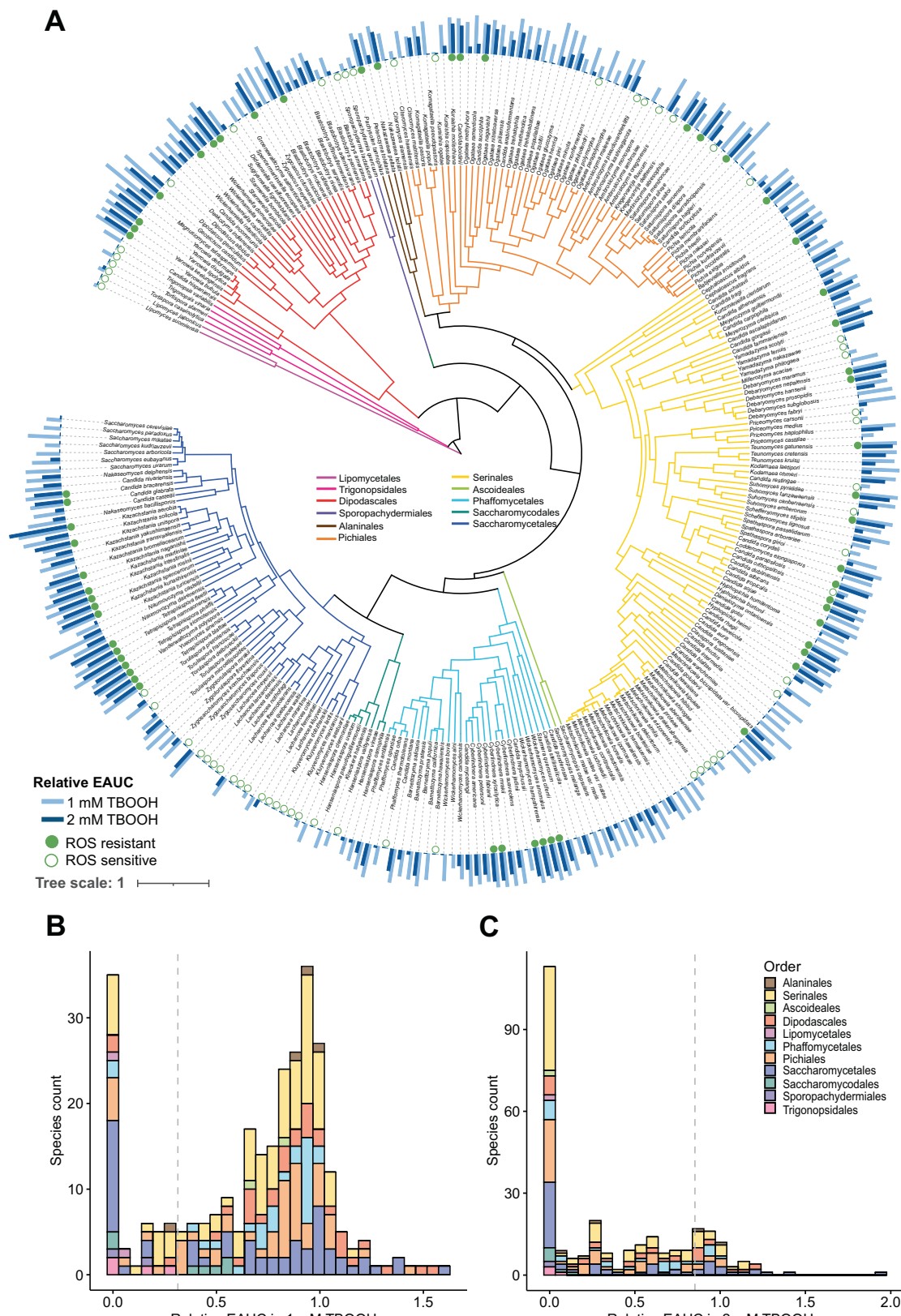

**Fig. 1 | The relative growth of yeast species grown in the presence of ROS. A** The empirical area under the curve (EAUC) values in 1 mM of the ROS-generator *tert*-butyl hydroperoxide (TBOOH, light blue) and 2 mM TBOOH (dark blue) are shown for each tested yeast species. The bar height represents the mean of three biological replicates. **B**, **C** Histograms showing the distribution of species' relative growth in the presence of TBOOH. **B** A histogram displaying the relative growth in 1 mM TBOOH with a vertical dashed line indicating the cutoff for the poorest-growing 20% of species, which were classified as ROS-sensitive (empty circles in **A**). **C** A histogram displaying the relative growth in 2 mM TBOOH with a vertical dashed line to highlight the cutoff for the best-growing 20% of species, which were classified as ROS-resistant (filled circles in **A**). Note that using both 1 mM and 2 mM TBOOH concentrations provided better contrasts to bin ROS-sensitive and ROS-resistant yeasts. Source data are provided as a Source Data file.

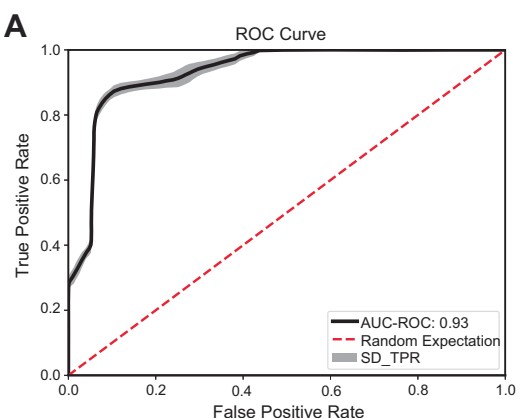

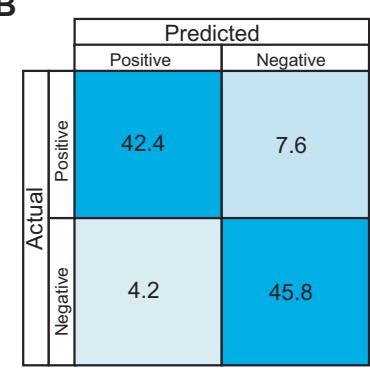

**Fig. 2 | Output metrics from the Random Forest classifier model. A** The area under the curve of the receiver operating characteristic (AUC-ROC) highlighting increasing identification of true positive classifications at varying thresholds of false positives. The standard deviation in the true positive rate (SD_TPR) is shown as a gray ribbon, whereas the black line represents the mean true positive rate. **B** The mean balanced confusion matrix from 100 model replicates, which highlights that model performance was generally accurate for both positive and negative classes. **C** A table summarizing the major model performance metrics for both the validation set of data, as well as the testing set that was not used to build the model; AUC-PR is the area under the curve of the precision-recall metric, and the F1 statistic is the harmonic mean of precision and recall. Source data are provided as a Source Data file.

the genes in the top 50 orthogroups from *S. cerevisiae* and *C. albicans* because these species have the most well-annotated genomes in our dataset. The GO term enrichment analysis revealed that predictive genes in both species were enriched in functions related to metal transport and the fungal cell wall. Enriched genes involved in cell wall organization and biogenesis included more specific terms that may be related to cell wall synthesis, such as protein glycosylation and mannosylation in *S. cerevisiae* and mannan biosynthetic processes in *C. albicans* (Fig. 3). Interestingly, the analysis of *C. albicans* genes uniquely revealed enrichment of GO terms related to host-pathogen interactions, including biological process involved in interaction with host, biofilm formation, and response to host defenses (Fig. 3C). Visualization of interactions between the identified genes using a Search Tool for the Retrieval of Interacting Genes/Proteins (STRING)[56] network revealed that cell wall-related genes were distributed across distinct hubs in both species (Fig. 3D, E), which highlights the importance of diverse features associated with cell wall biogenesis and organization. These results suggest that the cell wall and specific reductase gene families contribute to ROS resistance across diverse yeasts.

### Understanding the local influence of gene families on the classification model

Interpretation of the ML model allowed the identification of gene families predictive of ROS resistance. The top features of importance summarized by the RF algorithm were analyzed further to understand their local influences, or specifically, the quantitative contribution of each feature to the classification of individual species. The local influences were estimated using SHAP values (Supplementary Data 3), where positive and negative values indicate features contributing to ROS-resistant and ROS-sensitive predictions, respectively. By plotting SHAP values for each gene family on the phylogenetic tree as a heat map, we observed that the reductases had strong predictive power across the different orders (Fig. 4). Several of the top cell wall-related gene families had similar local influences across species, although there was some species-specific variation, particularly in the Phaffomycetales, Saccharomycetales, and Serinales. For example, many species in the genus *Metschnikowia* were sensitive to ROS and had negative SHAP values for several of the cell wall-related gene families, including those encoding GPI-anchored proteases, cell wall mannoproteins, and exo-1,3-β-glucanases. To further interpret these data, we also plotted the actual number of genes in each family against the

SHAP values for all species included in this model (Supplementary Fig. 2). This analysis revealed a clear and strong impact of gene family size on model classification for some gene families, such as the family encoding OYE reductase (OG0000030); for this family, any species with two or more copies had a positive SHAP value, but any species with one or no copies had a negative SHAP value. Other gene families had a less clear trend, including the family encoding mannosyltransferases (OG0000005), in which all species had positive SHAP values (Supplementary Fig. 2). In the orders Saccharomycetales and Serinales, species with more mannosyltransferase (*MNT*) genes generally had higher SHAP values. Altogether, SHAP values helped us interpret the local influence of features on ROS resistance class predictions. Since our most important gene families included many reductases and cell wall related enzymes, we used SHAP values to guide the selection of gene-species combinations for functional validation experiments of the *OYE* and *MNT* gene families. These validation experiments are ultimately vital for generating biological understanding and subsequent application of these results.

### The gene family encoding the reductase old yellow enzyme contributes to ROS resistance

The *OYE* gene family was one of the top features identified by our model, and we confirmed that ROS-resistant species had larger *OYE* gene families than ROS-sensitive species (Fig. 5A, B). When we considered all species screened and not only those included in our model, we found that gene family size was broadly correlated with growth in TBOOH (Fig. 5C). *K. lactis* had a negative SHAP value for *OYE*, which indicates that this gene family pushed *K. lactis* towards a ROS-sensitive classification. Since this species only has one copy of the gene encoding old yellow enzyme, *KYE1*, we tested whether increasing the copy number of *KYE1* conferred ROS resistance. We noted SHAP values increased with increasing *OYE* copy number (Supplementary Fig. 2), so we expressed an additional episomal copy of *KYE1* under the control of the moderately strong *K. lactis TEF1* promoter to mimic the addition of multiple *KYE1* copies. The overexpression of *KYE1* caused a general growth defect compared to the empty vector control, so we also used a *GFP*-expressing strain to control for the cellular demands of protein overproduction. The increased abundance of the resultant enzyme, Kye1, in the overexpression strains was confirmed by measuring the protein abundance using mass spectrometry (Supplementary Fig. 3). Under ROS stress induced by TBOOH, the strains overexpressing *KYE1* grew significantly better than the *GFP*-expressing controls. This result

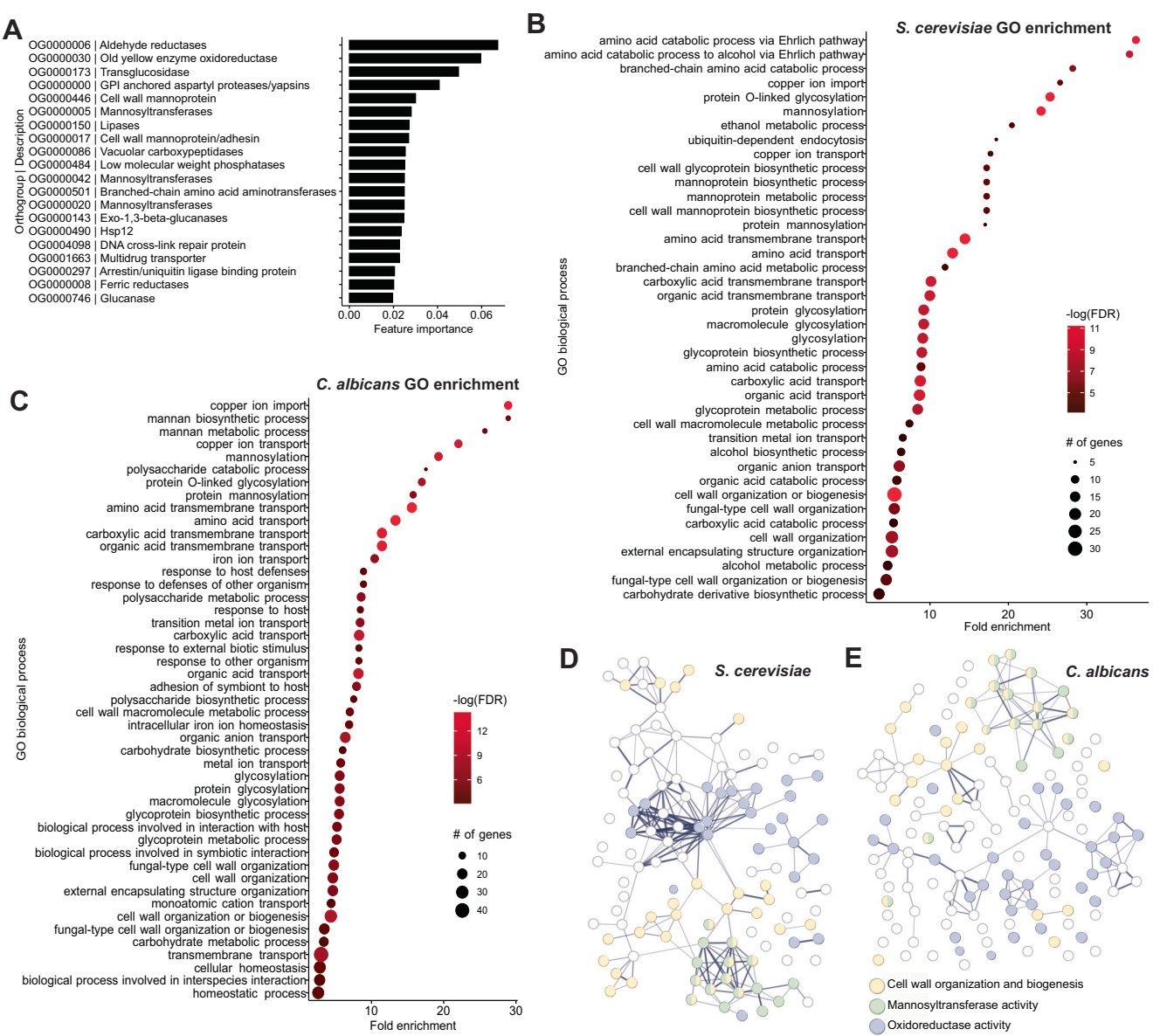

**Fig. 3 | Top features used to predict ROS resistance are enriched in genes involved in reductase activity and cell wall organization and biogenesis. A** The top 20 important features for the Random Forest classifier in their ranked order. For each gene family, the orthogroup (OG) and a general description of the gene are listed. **B**, **C** The enriched gene ontology (GO) terms based on the (**B**) *S. cerevisiae* and (**C**) *C. albicans* orthologs in the important features with a minimum of 5 genes and an FDR < 0.001 based on Fisher's exact test. **D**, **E** STRING (Search Tool for the Retrieval of Interacting Genes/Proteins) networks using the orthologs from (**D**) *S. cerevisiae* and (**E**) *C. albicans* highlighting the number and diversity of genes involved in (i) cell wall organization and biogenesis, (ii) mannosyltransferase activity, and (iii) oxidoreductase activity. Source data are provided as a Source Data file.

was observed both in a spot assay (Fig. 5D) and confirmed quantitatively using liquid growth curves ($p = 0.00038$, Fig. 5E). Interestingly, we noticed that the controls in the spot assays incubated at 30 °C were more sensitive to ROS stress compared to the room temperature incubation (Supplementary Fig. 4). To assess the general importance of old yellow enzyme in ROS resistance, we also grew these strains in the presence of two other ROS generators, hydrogen peroxide and menadione. Consistently, we found that additional episomal copies of *KYE1* significantly improved the growth of *K. lactis* compared to the *GFP*-expressing control in both $H_2O_2$ and menadione (Supplementary Fig. 5). Collectively, these results validate our model's prediction that increasing *OYE* gene family size improves ROS resistance.

**A role for *N*-mannosyltransferases in yeast ROS resistance**

The top 50 features of importance were enriched for genes related to cell wall organization and biogenesis. For functional validation, we considered gene families that showed a positive association between the number of genes and SHAP values (Supplementary Fig. 2) and were upregulated in *S. cerevisiae* upon $H_2O_2$ treatment in a previous study[57]. Therefore, we calculated SHAP values for *S. cerevisiae* (Supplementary Data 4) and found that the *MNT* gene family had a positive SHAP value. This result suggests that the large size of the gene family was contributing to its positive classification, which led us to hypothesize that reducing the size of this family would impair ROS resistance, thus making the *MNT* gene family a promising target for validation experiments. *MNT* gene family size was positively correlated with the relative growth in TBOOH across the yeasts assayed (Fig. 6A–C). To test causality, we made single deletions in *S. cerevisiae* of two *MNT* family genes, *KTR2* and *YUR1*, which are paralogs from the whole genome duplication[58], and tested their impacts on growth in the presence of ROS stress. We found subtle growth defects in both single mutants at high TBOOH concentrations, which prompted us to make a

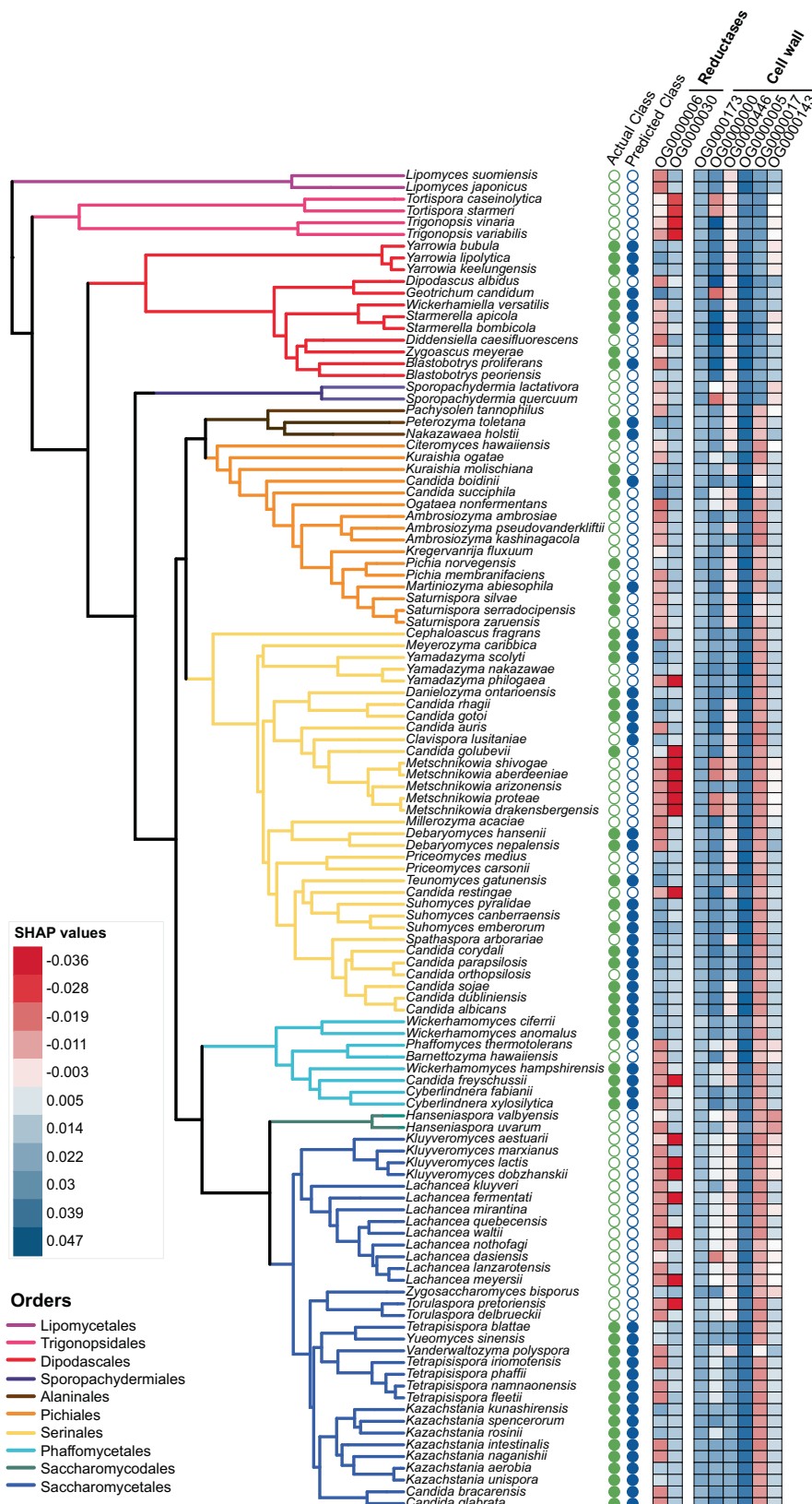

**Fig. 4 | Influence of gene families on classification across species using SHAP values.** A phylogenetic tree including the 114 species included in the ML model. The green filled circles indicate that the species was ROS-resistant (Actual Classification), whereas the blue filled circles indicated that the species was predicted as ROS-resistant by the model (Predicted Classification). The SHAP values for each species across the top gene families identified in the model related to reductase activity and cell wall organization are shown as a heat map. The red shading indicates a negative SHAP value, which contributes to classification as ROS-sensitive; whereas blue shading indicates a positive SHAP value, which contributes to classification as ROS-resistant. Source data are provided as a Source Data file.

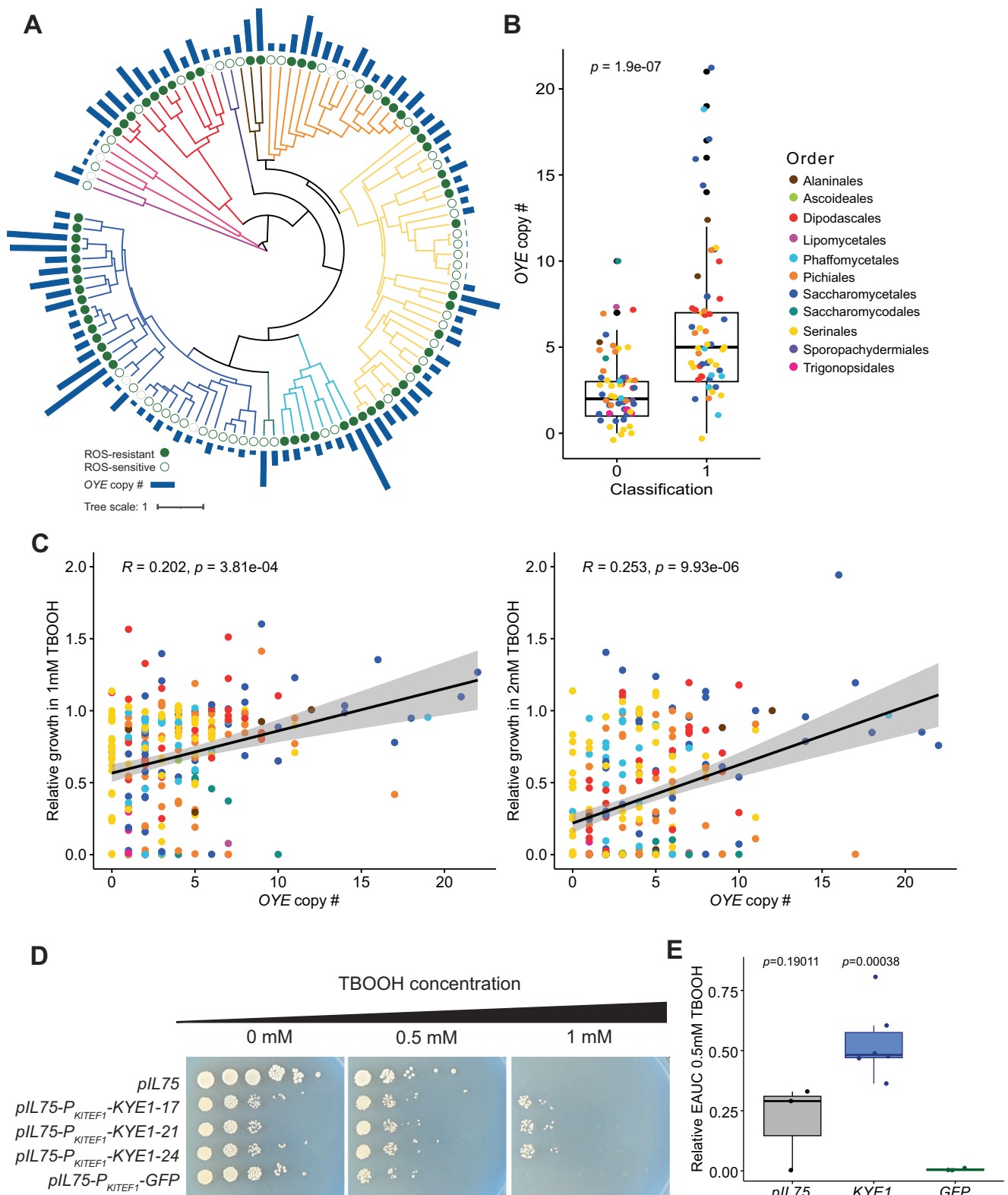

strain lacking both genes. In the double deletion mutants, we observed growth defects compared to the wild-type strain when grown in the presence of TBOOH (Fig. 6D, E). To confirm that there was an impact on the levels on mannosyltransferases in these strains, we used a proteomic approach. The abundance of Ktr2 was too low to be detected in any of our strains, including the wild-type strain. However, Yur1 peptides were only detected in the wild-type and single-knockout strain lacking only *KTR2*, showing that it was successfully deleted

(Supplementary Fig. 6). Consistent with the *K. lactis* experiment, we also found that all *S. cerevisiae* strains were more sensitive to ROS stress when grown at 30 °C than at room temperature. However, hypersensitivity of the *ktr2Δ yur1Δ* mutants was observed in the spot assays at both incubation temperatures (Supplementary Fig. 7). We also tested the sensitivity of these mutants to hydrogen peroxide and menadione as additional sources of oxidative stress. Interestingly, only the deletion mutant lacking *KTR2* grew significantly worse than the

**Fig. 5 | Increased old yellow enzyme (*OYE*) gene family size promotes ROS resistance in yeasts. A** A phylogenetic tree showing the variation in size of the *OYE* gene family in bars for the species included in the ML model. The classification of each species is indicated as a filled circle for ROS-resistant species and an empty circle for ROS-sensitive species. **B** A categorical comparison showing that ROS-resistant yeasts (Class 1) had, on average, more copies of *OYE* genes than ROS-sensitive yeasts (Class 0). The statistical significance was determined using a two-sided t-test, and the dots represent each species included in the model (*n* = 57 per class). **C** The correlation between the number of *OYE* genes in each species and their relative growth (empirical area under the curve, EAUC) at either 1 or 2 mM TBOOH as specified where the shaded area represents the 95% confidence interval. The *p*-values shown are based on t-tests of regression coefficients phylogenetically corrected with a generalized least squares model, and the *R* values are adjusted for phylogenetic effects. **D** A spot assay showing growth of the *K. lactis* strains

transformed with the following plasmids: an empty vector (*pIL75*), three independent transformants overexpressing the *OYE* ortholog *KYE1* (*pIL75-P_{KITEF1}-KYE1*) or *GFP* (*pIL75-P_{KITEF1}-GFP*) to control for the effect of protein overexpression. Plates were supplemented with varying concentrations of TBOOH as indicated and incubated at 30 °C for four days before imaging. Spot assays are representative images of three replicates. **E** The *K. lactis* strains were grown in liquid SC + MSG + G418 medium with or without 0.5 mM TBOOH in a 96-well plate format. The ROS resistance of these strains was compared using the EAUC in 0.5 mM TBOOH relative to no-TBOOH controls. The points in the boxplots represent biological replicates (*n* = 3 for *pIL75*, *n* = 6 for *KYE1* and *n* = 3 for *GFP*), and the *p*-values are based on two-sided t-tests relative to the *GFP* control. For all boxplots in this figure, the center line represents the media, the bounds of the boxes represent the interquartile range, and the whiskers represent the spread of the data. Source data are provided as a Source Data file.

wild-type strain in the presence of hydrogen peroxide, whereas all strains grew similarly in the presence of menadione (Supplementary Fig. 8). Overall, these results suggest that the importance of some mannosyltransferases is dependent on the source of oxidative stress. However, *KTR2* may be generally important for yeast growth in the presence of multiple peroxides.

## Discussion

Previous comparative screens of ROS resistance levels in yeasts have included only a small subset of species, often chosen for their relevance to biotechnology or human health[32,35–39]. In this study, we used an ML model to compare ROS resistance levels across yeast species with representation across most (11/12) taxonomic orders of the subphylum Saccharomycotina. We used growth data from two concentrations of TBOOH to classify yeasts as ROS-resistant or ROS-sensitive. This approach provided a unique opportunity to identify the broad importance of gene families predictive of ROS resistance, such as those encoding reductases and mannosyltransferases, which may be especially important in species that remain understudied and untapped for their potential contributions to biotechnology or unrecognized as potential emerging pathogens. Furthermore, using SHAP values to guide functional validation experiments allowed us to understand how the gene families we identified influence ROS resistance and to create a framework to interpret our ML model. These findings highlight potential targets that could be manipulated to increase the antifungal drug repertoire against yeast pathogens or to guide engineering strategies for the development of aerobic yeast cell factories for oleochemical or protein production.

The model's accuracy highlights the relevance of the sizes of specific gene families as a biological marker for oxidative stress resistance. Gene duplications and gene family expansions have long been recognized as important mechanisms of evolution[59]. Upon gene duplication, paralogs have several potential fates, including conservation, subfunctionalization, and neofunctionalization, which can increase the dosage of the duplicated gene or allow genes to function in a new context[60]. In yeasts, gene families related to stress responses are among those that have undergone expansion[61,62]. Therefore, we reasoned that using gene family sizes as features would allow for identification of large gene families that contribute to ROS resistance. Due to partial redundancy, these large gene families may have been particularly likely to be missed using traditional screening approaches with genome-wide knockout collections in a single species.

The most important features in our model were gene families encoding reductases, which have known importance in the oxidative stress response of genetic model systems. Here, we have generalized this genotype-phenotype connection across yeasts by showing that *OYE* gene family size was significantly correlated with ROS resistance across the species in our dataset, even after controlling for phylogenetic relationships. Mutants lacking the genes encoding OYE oxidoreductases are known to be hypersensitive to oxidative stress in *S.*

*cerevisiae*[63]. In one of the most sensitive species we tested, *K. lactis*, the sole *OYE* ortholog is upregulated at both the transcript and protein levels in response to $H_2O_2$, which suggests its importance in ROS resistance[29,64]. Our episomal overexpression of the *K. lactis OYE* ortholog functionally confirmed that increasing gene family size results in increased ROS resistance. Taken together, these findings suggest that *OYE* and the expansion of this gene family are broadly important for ROS resistance across yeast species.

Several of the most important features that we identified were cell wall-related genes. It has been established that there is cross-talk between the response to oxidative stress and the cell wall integrity pathway[65–69]. Cell wall changes associated with increased permeability to $H_2O_2$ are also likely to increase susceptibility to oxidative stress[70]. However, few studies in yeast have directly shown that mutants lacking cell wall biosynthetic enzymes have an impact on survival in oxidative stress. Interestingly, the protein *O*-mannosyltransferases Pmt1 and Pmt2 were found to support growth in TBOOH through their roles mannosylating a sensor of oxidative stress, Mtl1[71]. Outside of the yeasts, mannosyltransferases have been found to support ROS resistance in the entomopathogenic fungus in the subphylum Pezizomycotina, *Beauveria bassiana*[72]. We suspect that, in yeasts, the importance of large gene families contributing to ROS resistance, such as those encoding *N*-mannosyltransferases, has been overlooked due to redundancy, including in genetic model systems. This hypothesis is further supported by our functional validation data, which showed an impact on TBOOH resistance when multiple mannosyltransferase-encoding genes were deleted in *S. cerevisiae* (Fig. 6). The specific role of *N*-mannosyltransferases in yeast ROS resistance is novel, but it is consistent with prior observations that the cell wall integrity pathway is activated under ROS stress in *S. cerevisiae*[65]. The cell wall integrity pathway also coordinates with other stress response pathways, including the oxidative stress response through activation of sensors and crosstalk with the HOG pathway[73,74]. The cell wall of several pathogenic *Candida* species has also been shown to undergo morphological changes in the presence of $H_2O_2$, which provides further support for a relationship between the cell wall and yeast resistance to ROS[37]. Altogether, previous approaches showed that oxidative stress impacts cell wall integrity pathway activation but were unable to identify many important downstream cell wall-modifying enzymes. Through our comparative approach, we identified gene families encoding cell wall-modifying enzymes that are associated with ROS resistance and confirmed the importance of *N*-mannosyltransferases in resistance to TBOOH. However, future work is required to determine the exact mechanisms by which *N*-mannosyltransferases contribute to resistance against TBOOH and why the source of ROS stress impacts the importance of *N*-mannosyltransferases.

Although we validated the roles of two gene families predicted to be important in ROS resistance, this model has generated many hypotheses about other gene families that may be contributing to ROS resistance. We have calculated SHAP values for these gene-species

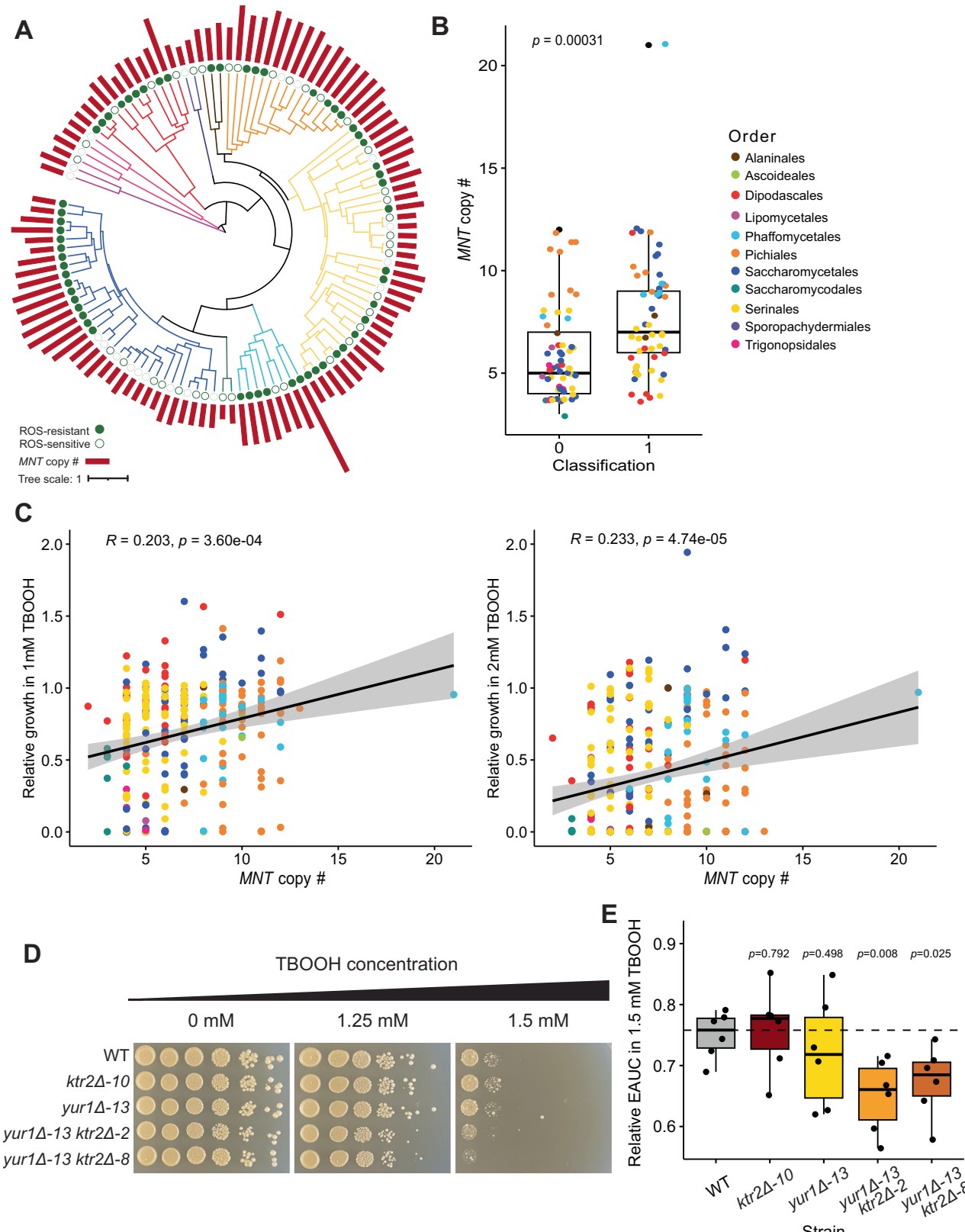

combinations (Supplementary Data 3), which could be used to guide future validation experiments and applications in specific yeasts. Furthermore, our validation experiments revealed that yeasts were more sensitive to TBOOH at 30 °C than at room temperature. This result is consistent with other reports that the oxidative stress response is required for survival at elevated temperatures[75–77] and highlights the

importance of exploring the interaction between ROS and temperature sensitivity, particularly in pathogenic yeasts, which face both in the context of host infection.

Herein, we have provided a framework for identifying gene families whose size is associated with a trait of interest. We have specifically identified gene families that are associated with ROS resistance

**Fig. 6 | The mannosyltransferase-encoding gene family contributes to ROS resistance in yeasts. A** A phylogenetic tree showing the variation in size of the mannosyltransferase-encoding (*MNT*) gene family represented as the length of the bars for the species included in the ML model. The classification of each species is indicated as a filled circle for ROS-resistant and an empty circle for ROS-sensitive. **B** A comparison of the number of *MNT* genes showing that ROS-resistant yeasts (Class 1) had, on average, larger gene families than ROS-sensitive yeasts (Class 0). The statistical significance was determined using a two-sided t-test, and the dots represent each species included in the model (*n* = 57 per class). **C** The correlation between the number of *MNT* genes in each species and their relative growth (empirical area under the curve, EAUC) at either 1 or 2 mM TBOOH as specified where the shaded area represents the 95% confidence interval. The *p*-values shown are based on t-tests of regression coefficients phylogenetically corrected with a

generalized least squares model, and the *R* values are adjusted for phylogeny. **D** A spot assay showing growth of the *S. cerevisiae MNT* deletion strains at varying concentrations of TBOOH as indicated. Plates were incubated at 30 °C for three days before imaging. Spot assays are representative images of three replicates. **E** A categorical comparison of the impact of ROS stress on the growth of each strain in liquid medium based on the empirical area under the curve (EAUC) of growth curves in 1.5 mM TBOOH relative to no-TBOOH controls. The points in the boxplots represent six biological replicates per strain, the boxes represent the interquartile range, the dashed line indicates the median for the wild-type (WT) strain, and the *p*-values are based on two-sided t-tests relative to the WT control. For all boxplots in this figure, the center line represents the media, the bounds of the boxes represent the interquartile range, and the whiskers represent the spread of the data. Source data are provided as a Source Data file.

and used SHAP values to inform the design of validation experiments. This approach could be applied to other yeast species where the manipulation of ROS resistance may be desirable. For example, in oleaginous yeasts, including *Y. lipolytica*, ROS-mitigating modifications have been shown to increase lipid titers[21]. Oxidative bursts are used by macrophages to kill microbes in the phagosome, so the ability of pathogenic yeasts to avoid or survive these assaults are crucial to their virulence[24,25]. Thus, our data suggest potential targets for antifungal drug development to potentiate susceptibility to ROS produced by macrophages. These data also suggest possible therapeutic approaches combining existing drugs that target the cell wall and drugs that are known to induce ROS generation[78].

As ROS is a common stressor encountered through microbial interactions with hosts, other microbes, or the environment[20], systematically investigating the diverse resistance mechanisms of yeasts has far-reaching implications for applications in human health and microbial bioproduction. This work used a rich genomic dataset to identify previously overlooked gene families predictive of ROS resistance, such as the *N*-mannosyltransferase-encoding family, and it underscores the complexity of ROS resistance because several different gene families contribute to correct predictions of ROS resistance across yeast species. We also provide a comparative framework to circumvent the challenge of redundancy and identify gene family expansions involved in trait acquisition across yeasts. With ever-increasing amounts of available data, ML models can be used for the exploration of traditionally cryptic aspects of trait diversity across yeast species, ultimately guiding yeast manipulation to benefit humanity.

## Methods

### Growth conditions and species used
Resistance to ROS in 285 yeast species within the subphylum Saccharomycotina was measured based on quantitative growth curves (Supplementary Data 1). All yeasts were revived from glycerol stocks and routinely grown at room temperature (22 °C) on yeast peptone dextrose (YPD) agar (US Biologicals). For phenotypic screening, all species were inoculated in 96-well plates in YPD broth and grown for three days before being diluted 10-fold into sterile water and pinned into experimental plates containing 200 μl synthetic complete medium with 2% dextrose (SCD, US Biologicals) supplemented with 0, 1, or 2 mM *tert*-butyl hydroperoxide (TBOOH, Sigma, 458139). TBOOH was chosen as the oxidative stress agent since our yeast species have variable lag phases, and $H_2O_2$ decomposes rapidly, particularly in the presence of sugars[79,80]. If we had chosen $H_2O_2$ for the screen, some slow growing yeasts would not have been exposed to the same effective concentration of oxidizing agent during their exponential growth phase. Nunc 96-well Edge™ plates (ThermoFisher) were used, and all four edges were filled with 3 ml of sterile water to reduce the impact of evaporation in edge wells. Plates were incubated at room temperature (22 °C) and read every hour at $OD_{600nm}$ for 168 h using a SPECTROstar Omega Plate Reader equipped with a Microplate Stacker (BMG

LabTech). Whole plate outputs were analyzed using the *growthcurver* package[81] in *R* v4.2.2. Relative growth amounts were determined based on the empirical area under the curve (EAUC) in the presence of TBOOH relative to the EAUC in the no-treatment control for each replicate. The averages of three biological replicates performed on different days were used in subsequent analyses. The difference in average ROS resistance between orders was determined using an Analysis of Variance (ANOVA) in *R*.

### Machine learning classification model
Species were classified based on their relative growth in the presence of TBOOH as follows: the 20% of species that grew the poorest in 1 mM TBOOH were classified as ROS-sensitive (Class 0), and the 20% of species that grew the best in 2 mM TBOOH were classified as ROS-resistant (Class 1) for a total of 114 species as instances. This class column was combined with the orthogroup matrix containing counts for all gene families across all species (Supplementary Data 5), which was previously generated for the subphylum Saccharomycotina[3] using OrthoFinder[82]. The resulting input data matrix consisted of 114 species in rows, 72,380 unique orthogroups in columns, and one class column (Supplementary Data 5). Before training the model, 10% of instances were randomly held out for testing with the condition that an equal number of positive and negative instances were chosen for removal to avoid biasing the model. We used an ML pipeline with automated tuning of the following hyperparameters: the number of estimators, the maximum tree depth, and the maximum features used at each branch point (https://github.com/ShiuLab/ML-Pipeline)[40]. To choose the most predictive features from the orthogroup matrix, we performed feature selection using the Random Forest (RF) algorithm, specifying the number of features to select as 50 and the type as classification. The 50 most important features were selected based on Gini importance, which represents the normalized contribution of each feature to the purity of child nodes across all trees in each model. Using these 50 features, we ran an RF classification algorithm using a full grid search to determine the best combination of the number of estimators to use, the number of features to consider when looking for the best split, and maximum tree depth. A total of 500 trees with a maximum depth of 10 were used for each model. For each of 100 model replicates, a 10-fold cross-validation scheme was applied to assess the model's generalization ability. Model performance was evaluated using the tradeoff between precision and recall (F1) statistics, the area under the curve of the receiver operating characteristic (AUC-ROC), and the area under the curve of the precision-recall (AUC-PR). The mean balanced confusion matrix for the validation set was calculated by averaging the confusion matrices from 100 independent model training runs and the model was trained on unique subsets of the data for each iteration. Metrics were generated for the validation dataset, which includes all species used for training, as well as for the testing set, which includes species not previously seen by the model to assess the model's generalization ability.

## Gene Ontology term enrichment of important features

The top 50 features of importance selected by the model were used for a Gene Ontology (GO)[83] term enrichment analysis to identify genes broadly related to ROS resistance across all species in the subphylum. For each orthogroup identified as an important feature, we retrieved the *S. cerevisiae* and *C. albicans* orthologs and conducted a GO term enrichment analysis for each species in pantherdb.org[84] (accessed April 4, 2024). Respective background genes for each species were extracted from the orthogroup matrix to ensure only the genes included in our dataset were considered. The significance was determined using Fisher's exact test and a false discovery rate (FDR) threshold of 0.001 was used for visualization to highlight the most significant enrichments. Fold-enrichment was calculated by pantherdb as the number of genes in our list divided by the expected number from the background genes we provided. The genes identified were further visualized in a network using string-db.org[56] (accessed April 4, 2024).

## Model interpretation using Shapley additive explanations

To understand the direction of influence that each orthogroup has on the classification of individual species, SHAP values were estimated by the Tree Explainer using Python's SHAP package v0.28.5[85]. SHAP values were plotted onto a phylogenetic tree using iTOL v6[86]. The relationships between SHAP values and the number of genes within the most predictive orthogroups were also investigated by plotting these values using the *R* package *ggpubr*[87].

## Comparing gene family size and reactive oxygen species resistance for top features

The numbers of orthologs of the old yellow enzyme-encoding (OG0000030) and mannosyltransferase-encoding (OG0000005) gene families from each species were retrieved from the orthogroup matrix. These values were plotted outside a pruned phylogenetic tree to show the number of orthologs in each species included in the model using iTOL v6[86]. The correlations between relative growth in TBOOH and the number of orthologs were visualized using *ggpubr*[87], and the correlations were corrected for phylogenetic signal using a phylogenetic generalized least squares (PGLS) model in *caper*[88].

## Overexpression of old yellow enzyme reductase in *Kluyveromyces lactis*

*K. lactis KYE1* encodes OYE and is the species' sole ortholog to the *S. cerevisiae* S288C reference genes *OYE2* and *OYE3*, as well as the original *OYE1* from hybrid lager yeast[89,90]. Using the primers listed in Supplementary Data 6, plasmids were constructed using HiFi assembly master mix (NEB) to ligate into the amplified *pIL75* vector backbone, which encodes G418 resistance[91]: 1) a green fluorescent protein (GFP)-encoding control gene from pYTK047[92] or 2) the *KYE1* gene amplified from *K. lactis* genomic DNA. Both expression plasmids, as well as an empty vector control, were transformed into *K. lactis*, which retained the native copy of *KYE1*, using the LiAc/heat shock method as previously described[93]. Transformants were plated onto YPD + G418 to select for strains with a plasmid and streaked out for single colonies on fresh YPD + G418. The plasmid expressing *KYE1* was verified by whole plasmid sequencing performed by Plasmidsaurus using Oxford Nanopore Technology with custom analysis and annotation. The *GFP*-expressing plasmid was confirmed using colony PCR, restriction enzyme digest with *Hind*III, and by checking fluorescence after transforming into *K. lactis*.

All strains were grown in SC medium made with 1% monosodium glutamate as the nitrogen source supplemented with 200 µg/ml G418 sulfate at 30 °C overnight in biological triplicate. Harvested cells were washed once with sterile water and normalized to $OD_{600nm} = 1$. Liquid growth assays were conducted in 96-well plates in 200 µl SC + MSG + G418 (supplemented with or without 0.5 mM TBOOH)

inoculated at $OD_{600nm} = 0.1$. Plates were placed into the SPECTROstar Omega Plate Reader with a Microplate Stacker and read every hour at $OD_{600nm}$ for 72 h. To assess growth on solid medium, ten-fold serial dilutions of each strain normalized to $OD_{600nm} = 5$ were spotted onto SC + MSG + G418 agar plates supplemented with four concentrations of TBOOH: 0.25, 0.5, 1, and 2 mM, as well as a no-TBOOH control. Solid plates were incubated at room temperature (22 °C) to reflect the conditions used in the original screen or at 30 °C, which is the normal laboratory growth temperature for *K. lactis*, for four days before imaging.

## Deletion of cell wall-related genes in *Saccharomyces cerevisiae*

The genes *KTR2* and *YUR1* from the mannosyltransferase-encoding gene family were individually deleted from *S. cerevisiae* strain S288C by replacing the open reading frame with a *kanMX*- resistance cassette. The *kanMX*-resistance cassette was amplified from pUG6[94] using primers with overhangs to direct homologous recombination (Supplementary Data 6). To delete *KTR2* from the *YUR1* single-deletion mutant, the *KTR2* gene was replaced with a *natMX*- resistance cassette amplified from pAG25[95] and reamplified using primers with overhangs to direct homologous recombination (Supplementary Data 6). Transformants were plated onto YPD + G418 + 100 µg/ml Nat to select for double knockouts and streaked out for single colonies on fresh YPD + G418 + Nat. The strains generated are listed in Supplementary Data 7. Successful gene replacements were verified using colony PCR with primers listed in Supplementary Data 6.

Strains were grown in SC medium at 30 °C overnight (24 h) in six biological replicates per strain. All cells were harvested, washed once with sterile water, and normalized to $OD_{600nm} = 1$. For liquid growth assays, strains were grown in 96-well plates inoculated at $OD_{600nm} = 0.1$ into SC supplemented with 1.5 mM TBOOH. The position of strains in these plates were randomized to control for any edge effects. Plates were incubated at room temperature and read using a SPECTROstar Omega Plate Reader with a Microplate Stacker and read every hour at $OD_{600nm}$ for 72 h (BMG LabTech). For assessment of growth on solid medium, ten-fold serial dilutions of each strain normalized to $OD_{600nm} = 5$ were spotted onto SC agar plates (supplemented with or without TBOOH). Plates were incubated at room temperature (22 °C), which was the temperature used for our high-throughput screening, or 30 °C, which is the temperature for routine laboratory growth of *S. cerevisiae*, for three days before imaging.

## Impact of additional oxidative stress agents

For the *K. lactis* experiments investigating the old yellow enzyme, strains were grown in SC medium made with 1% monosodium glutamate as the nitrogen source supplemented with 200 µg/ml G418 sulfate at 30 °C overnight in biological quadruplicate. Harvested cells were washed once with sterile water and normalized to $OD_{600nm} = 1$. Liquid growth assays were conducted in 96-well plates in 200 µl SC + MSG + G418 (supplemented with 2 mM $H_2O_2$ or 30 µM menadione) inoculated at $OD_{600nm} = 0.1$. Plates were placed into the SPECTROstar Omega Plate Reader with a Microplate Stacker and read every hour at $OD_{600nm}$ for 72 h.

For the *S. cerevisiae* experiments investigating mannosyltransferases, strains were grown in SC medium at 30 °C overnight in biological quadruplicate. Harvested cells were washed once with sterile water and normalized to $OD_{600nm} = 1$. Liquid growth assays were conducted in 96-well plates in 200 µl SC (supplemented with 2 mM $H_2O_2$ or 50 µM menadione) inoculated at $OD_{600nm} = 0.1$. Plates were placed into the SPECTROstar Omega Plate Reader with a Microplate Stacker and read every hour at $OD_{600nm}$ for 48 h.

## Growth curve analyses

Liquid growth curves were analyzed using the *R* package *growthcurver*[81], and growth curves were visualized using the *R* package

*ggplot*[96]. Relative EAUC was calculated for each strain grown in TBOOH compared to the no-TBOOH control. Average relative EAUC values were calculated across biological replicates, and significance was determined using a two-sided t-test for pairwise analyses and a Pearson's correlation for correlative analyses.

## Proteomic sample preparation

Overnight cultures were grown in biological triplicate under the same pre-growth conditions used to test sensitivity to TBOOH, and the cells from 1 ml of culture were collected by centrifugation and snap frozen in a dry ice ethanol bath. Samples were prepared as described previously[97]. Briefly, cell pellets were thawed on ice before resuspension in Lysis Buffer consisting of 8 M urea, 10 mM TCEP, 40 mM CAA, and 100 mM TRIS. Samples were bath-sonicated for 5 min at room temperature (Thermo Fisher, FS60H). Protein concentration was determined by BCA (Pierce) before transferring 100 μg to new tubes. Methanol was added to 90% final volume, and samples were then vortexed and centrifuged at 12,000 x g for 5 min at ambient temperature. The supernatant was removed, and 50 μL Lysis Buffer was added to resuspend the pellet. Samples were digested overnight at ambient temperature with trypsin (Promega), using a 1:50 enzyme:protein ratio. The following day, samples were acidified with TFA to reach a pH of 2. Acidified peptides were desalted with 10 mg Strata X desalting columns (Phenomenex), following vendor recommended protocol. Eluted peptides were dried by vacuum centrifugation.

## Liquid chromatography mass spectrometry (LC-MS) proteomic analysis

Peptides were resuspended in 50 μL 0.2% formic acid and were quantified by UV/Vis using a Nanodrop One (1 Abs = 1 mg/mL, Thermo Fisher). For liquid chromatography mass spectrometry (LC-MS) analysis, 250 μg of peptide was loaded onto a house-packed ~45 cm capillary column (75–360 μm inner-outer diameter bare-fused silica shells with laser-pulled electrospray emitter tip; made in house)[98] filled with 1.7 μm, 130 Å pore size, Bridged Ethylene Hybrid (BEH) C18 particles (Waters), maintained at 50 °C inside an in-house made heater. The elution gradient was delivered by a Vanquish Neo nanoLC system (Thermo Fisher) at a flow rate of 0.3 μL/min over a 30-min active gradient (from 10%-54% Mobile Phase B). Mobile Phase A consisted of 0.2% formic acid in water, and Mobile Phase B consisted of 80% acetonitrile and 0.2% formic acid in water.

Mass spectra were acquired in a data independent analysis (DIA) approach using an Orbitrap Astral mass spectrometer (Thermo Fisher) in positive mode with settings as follows: Orbitrap MS1 scans were performed at 240k resolution every 0.6 s over 380–980 $m/z$ with normalized AGC target of 250%, maximum injection time of 50 ms, and an RF lens setting of 40%. MS2 scans were performed in the Astral mass analyzer with a precursor isolation range of 380–980 $m/z$ divided into 2 $m/z$ isolation windows resulting in 300 scans per cycle. Normalized AGC target was set to 500% with a 3 ms maximum injection time. HCD collision energy was set to 25% with a default charge state of +2, and expected peak width was set to 5 s.

## Proteomic data processing

The resulting LC-MS data were processed using separate dDIA searches for each organism on Spectronaut (v 19.5.241126.62635, Biognosys). Searches used the *S. cerevisiae* and *K. lactis* reference proteomes (both Swiss-Prot and TrEMBL) downloaded from UniProt on 16-Dec-2024. Default search settings were used with the following changes: precursor Qvalue cutoff, precursor PEP cutoff, protein Qvalue cutoff (experiment), protein Qvalue cutoff (Run), and protein PEP cutoff were all set to 0.01[99]. Only proteotypic peptides were used for quantification, and "all matching proteins" was used as the inference algorithm. The searched data was filtered by removing protein groups observed in fewer than 50% of replicates in at least one condition and log$_2$ transformed[100].

## Reporting summary

Further information on research design is available in the Nature Portfolio Reporting Summary linked to this article.

## Data availability

The MS data generated for measuring protein levels in this study have been deposited in the MassIVE repository (https://massive.ucsd.edu/ProteoSAFe/static/massive.jsp) with the dataset identifier MSV000096915. Upon request from the corresponding author, all engineered strains are available under the Uniform Biological Material Transfer Agreement or another mutually agreeable material transfer agreement. Source data are provided with this paper.

## Code availability

Scripts for building the ML model, model analysis, and generating figures associated with performance metrics are available on GitHub (https://github.com/katarinaaranguiz03/Yeast_ML_ROS)[101]. Versions of all dependencies and packages are also listed in the provided YAML file. A more general workshop and pipeline are also available on GitHub (https://github.com/ShiuLab/ML-Pipeline).

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

## Acknowledgements

The authors thank members of the Hittinger Lab and Y1000+ Project (http://y1000plus.org) team members for helpful discussions. This project was supported by the National Science Foundation under Grants No. DEB-2110403 (to C.T.H.); DEB-2110404 (to A.R.); IOS-2107215, MCB-2210431, and IOS-2218206 (to S.H.S.); in part by the Great Lakes Bioenergy Research Center, U.S. Department of Energy, Office of Science, Biological and Environmental Research Program under Award Number DE-SC0018409 (of which C.T.H., S.H.S and J.J.C. are co-investigators); and the National Institute of Food and Agriculture, United States Department of Agriculture, Hatch project 7005101 (to C.T.H.). C.T.H. is an H. I. Romnes Faculty Fellow, supported by the Office of the Vice Chancellor for Research and Graduate Education with funding from the Wisconsin Alumni Research Foundation. Research in A.R.'s lab is also supported by the National Institutes of Health/National Institute of Allergy and Infectious Diseases (R01 AI153356) and the Burroughs Wellcome Fund. Research in J.J.C.'s lab is also supported by the National Institutes of Health grant R35GM118110. L.C.H. was supported by a Natural Sciences and Engineering Research Council of Canada (NSERC) postdoctoral fellowship. K.A. was supported by a Sophomore Research Fellowship, Biochemistry Fellowship, and Hilldale Fellowship from the University of Wisconsin-Madison. D.J. is supported by the National Institute of General Medical Sciences of the National Institutes of Health under Award Number T32GM135066 and by the National Science Foundation Graduate Research Fellowship program under Grant Number 2137424.

## Author contributions

K.A., L.C.H., and C.T.H. conceptualized this project. K.A. and L.C.H. developed methodology with the input and guidance of K.S.A., S.H.S., A.R., and C.T.H. D.J. and K.A.O. performed and analyzed proteomic data. R.L.W. generated plasmids and provided expertise on old yellow enzyme. K.A., L.C.H., and L.E. performed all experiments and analyzed data. K.A., L.C.H., and C.T.H. wrote the manuscript, and all authors reviewed, edited, and approved the manuscript. C.T.H., A.R., S.H.S., J.J.C., L.C.H., and K.A. acquired funding for this project. L.C.H. and C.T.H. mentored K.A. L.E. was mentored by L.C.H. and C.T.H. during his time in Biological Interactions Research Experience for Undergraduates.

## Competing interests

A.R. is a scientific consultant for LifeMine Therapeutics, Inc. The other authors declare no competing interests.
