## [Transparent Peer Review file · Nature Communications]

Machine learning reveals genes impacting oxidative stress resistance across yeasts

Corresponding Author: Professor Chris Hittinger

Version 0:

Reviewer comments:

Reviewer #1

(Remarks to the Author)

In this manuscript, Aranguiz et al. use a Random Forest ML algorithm to identify gene families that predict high ROS resistance across diverse yeast species spanning a large phylogenetic space. The authors measured levels of ROS resistance via plate reader growth analysis of 285 different yeast species on two different concentrations of tert-butyl hydroperoxide, and then used the 20th percentile tails of the distribution to define species as ROS-sensitive or resistant. They applied a ML algorithm that used the phenotype data and previously generated orthogroup assignments across the species to identify gene families whose size correlated with ROS resistance. Two of the top families included reductases and genes involved in cell wall synthesis, and the authors validated the predictive value of the model by over-expression of an old yellow enzyme reductase in an ROS-sensitive species, as well as deletion of mannosyltransferase-encoding genes in ROS-resistant *Saccharomyces cerevisiae*.

I think this paper very nicely illustrates how ML models can leverage comparative genomics and phenotype data to make predictions about genotype-phenotype relationships. I think this paper is of broad interest, in addition to being well written with a clear and concise logic throughout. I do not have any major issues, and have only a few minor comments.

I should point out that my expertise is in fungal genetics/genomics and stress biology, and not ML algorithms. As such, I could conceivably miss any issues with the ML model. With that caveat in mind, the AUC-ROC and F1 scores both point to a model that has reasonably strong predictive quality for the testing dataset. Based on the held out test data, it appears that the model has an ~10% false positive rate and an ~15% false negative rate, which from my perspective seems pretty good. I do wonder how certain choices may influence model performance, such as classifying ROS-sensitive and resistant species based on the 20th percentile tails instead of the 10th percentile (or 30th), or if phenotype data was entered as relative EAUC values instead of classifications. Another minor caveat of the model in my mind is that the authors leveraged genes that were known to be upregulated by oxidative stress when choosing candidates for over-expression and knock-out experiments. That's a sound strategy that turned out to be fruitful, but it seems like that could be included in the model to either rank genes within a family, or to even be used on the front end to help rank the families beyond family size. I bring these things up not necessarily for changing the model for this particular paper, but because I hope the authors might consider further iterations down the road. This seems like something that could be a useful standalone software package for the fungal community, and I also hope that the authors consider releasing the model in more accessible format beyond what is on the github repository in the future. I can imagine a package that allows users to input phenotype data and then outputs top features and their SHAP values would be widely used. Please take this overall comment as 'food for future thought' and not something that I expect be added to this paper.

Minor comments:

1) Line 141-142: I think it's important when thinking about variation in ROS resistance among pathogens that their in vivo ROS resistance may be different from their ROS resistance under laboratory conditions, as they may transcriptionally induce different ROS defenses in vivo.

2) Were the strains used for assaying ROS resistance the exact same genotype as the sequenced isolate? I mention this because gene number can vary across strains within a species, and variation in the number of ENA genes in *S. cerevisiae* is notably linked to variation in salt resistance across strains.

3) It would be useful to compare family predictions to genes with known ROS sensitivity from deletion library screens in *S. cerevisiae* (e.g., DOI 10.1016/j.freeradbiomed.2007.12.008). The authors have a great example of redundancy masking the effects of KTR2 and YUR1 single deletions, but it would be interesting to know if there are families that are likely systematically underrepresented in deletion library screens.

4) Discussion: paragraph starting at 385. I do not think it is entirely fair to say that cell-wall related genes have not been previously identified as important for yeast ROS resistance. There are a number of papers that have found connections between the CWI pathway and oxidative stress (DOIs 10.1128/EC.00245-10, 10.1074/jbc.M411062200, 10.1016/j.fgb.2018.03.002), as well as potentially direct connections between cell wall protecting against oxidative stress resistance (DOIs 10.1128/ec.2.5.1018-1024.2003, 10.1016/j.febslet.2004.10.090, 10.1074/jbc.M411062200). There seem to also be connections to mannosylation and oxidative stress resistance (DOIs 10.1016/j.fgb.2012.08.005, 10.1016/j.bbrc.2006.12.012). It is fair to say that the involvement of KTR2 and YUR1 is a novel validation of the prediction, but I think it is somewhat overstating the case that cell wall and mannosylation involvement in oxidative stress resistance is a novel connection.

(Remarks on code availability)

I did not run the code, but I did look at the code and documentation, and there appears to be enough information for someone moderately versed in python to run the code.

Reviewer #2

(Remarks to the Author)

“Machine learning reveals genes impacting oxidative stress resistance across yeasts” by Aranguiz et al. describes tert-butyl hydroperoxide tolerance across 285 species of yeasts. This is an impressive phenotypic analysis across all the phylogenetic tree of single cell fungi. Using machine learning the authors were able to pinpoint the role of gene families such Old Yellow Enzyme and mannosyltransferases in such phenotype. The experimental validation of the expected families indicates that this approach is valid to identify new players in any phenotype with potential biotechnological or clinical implications, particularly those involved families of redundant genes that may be ignored in systematic deletion analysis. However, the choice of t-BOOH as oxidative stress inductor is not obvious and it has to be explained regarding molecular mechanisms or experimental conditions. Hydrogen peroxide would be the most obvious choice. It is indicated three times in the text talking about oxidative stress induced genes and tolerance mechanisms (308, 380 and 400). This is a molecule all cells produce naturally, and it is known in *S. cerevisiae* that different peroxiredoxins deal with H₂O₂ and alkyl peroxides, and the signature of the stress response is different (for instance see Iwai K, Naganuma A, Kuge S, 2010. Peroxiredoxin Ahp1 acts as a receptor for alkylhydroperoxides to induce disulfide bond formation in the Cad1 transcription factor. *J. Biol. Chem.* 285: 10597–10604). A superoxide generator like menadione would mimic better the burst in superoxide anions generated by macrophages to deal with pathogenic yeast. The genetic modifications described in Figures 5 and 6 that were tested for tert-butyl hydroperoxide tolerance have to prove their role in hydrogen peroxide and other stressors to prove that the ROS resistance that was indicated in the abstract (where t-BOOH was not indicated) is real. Besides, the impact of *S. cerevisiae* mannosyltransferases deletion is clear, but the mutant was not “hypersensitive” as indicated in the Abstract (line 40) or had “strong impact on ROS resistance” as mentioned in line 392.

Minor points

Explain what “FAD transport” in line 195 is referred to.

OYE enzymes have been involved in the reduction of citral to citronellal, and event of biotechnological relevance. Are the OYE genes also included in the category “aldehyde reductases of Figure 3A.

(Remarks on code availability)

Reviewer #3

(Remarks to the Author)

1. Summary:

This study explores the use of gene family size as a feature for machine learning models to predict phenotypic traits in yeast species, specifically their sensitivity or resistance to reactive oxygen species (ROS). The authors developed a Random Forest (RF) classifier using gene family size to predict ROS resistance across over 100 yeast species. The model highlighted reductase and cell wall-related gene families as the most predictive features. The study's key biological finding is that gene families encoding reductases were among the most predictive features, and the top 50 features were enriched for cell wall-related gene families. Experimental validation confirmed that increasing the copy number of genes encoding old yellow enzyme (OYE) increased ROS resistance in *Kluyveromyces lactis*, while deleting mannosyltransferase genes in *Saccharomyces cerevisiae* decreased ROS resistance. Overall, this study demonstrates a novel approach to connecting gene family size with trait variability in yeast, with potential applications in both biotechnology and antifungal development.

2. Key strengths

The manuscript is well-written, and the experimental work is thorough, especially the ROS resistance assays involving over 100 yeast species. The integration of machine learning with comparative genomics is a powerful approach that provides new insights into the genetic basis of ROS resistance. The figures, despite being dense and with fonts that will be too small

to read in print (but okay when zoomed in), are still visually clear and convey the experimental scope and effort behind the work.

Although the machine-learning algorithms used in this study is not new - random forest with SHAP is one of the most frequently used Python packages - the novelty of using gene family size as an input feature for a machine learning model is compelling, and the study demonstrates the utility of this approach for identifying key genetic components related to stress resistance in yeast. The findings have practical applications in biotechnology (e.g., improving yeast strains for industrial production) and clinical settings (e.g., targeting pathogenic yeasts). The validation experiments, particularly the overexpression and deletion studies, solidify the model's predictions and provide valuable biological insights.

3. Overall recommendation:

I recommend publication with minor revisions. But I do have some recommendations for the authors. The authors can decide whether to address these if they don't push the study beyond its current scope. I don't believe any of these points require additional experiments but might require some rewriting and further analyses of their existing data.

4. Major points:

1. L167-168: RF Feature Selection Algorithm Clarification:

The description of the "RF feature selection algorithm" is somewhat unclear. It would benefit from additional details about the feature selection process. Did the authors rely on SHAP values from the RF model to rank the features? Also, while gene family size is a primary input feature, how many total features were used in the model, and what was the architecture of the RF classifier (e.g., the number of trees, the depth of trees)? A better explanation of the complexity and interpretability of the RF model would help readers understand how the gene family size relates to the model's predictions. Adding this information would clarify the modeling process and emphasize the significance of the selected features.

2. Clarity of RF Model Setup (related to above):

The manuscript could benefit from a clearer explanation of how gene family size was integrated into the model and how feature selection was performed. Was there any pre-filtering of gene families before they were included as features, or were all gene families across all species considered in the model?

3. Use of Random Forest (RF) model:

The application of RF classifier in this study was well-suited for identifying gene families predictive of ROS resistance. However, the authors could have explored additional machine learning models to compare performance, such as gradient-boosting machines or support vector machines. RF is a solid choice and it is one that is perhaps more friendly and easier to use for those new to machine learning, making this work broadly appealing to readers. But a broader exploration of other machine-learning models might reveal further insights. The authors might consider justifying why they chose RF instead of other ML models. If they explored other ML models, explain how they fared, in terms of predictability and interpretability, compared to the RF model.

4. Fig. 2C: Include Recall Value:

The manuscript provides metrics such as the F1 score and precision for the RF classifier but omits the recall value. Including the recall value would give a more complete picture of the model's performance, especially in understanding how well it identifies true positives. Additionally, showing the confusion matrix would help readers grasp the balance between false positives and false negatives in the classification model.

5. Fig. 3A: Justification for Focus on Top Features:

The feature importance plot in Fig. 3A shows that the top two features (gene families encoding reductases) have an importance value of around 0.06, while many other features in the top 50 have values around 0.02—a difference of only about 3-fold. This suggests that many features contribute to the model's performance. While the authors focus on the top two features, it would be fairer to acknowledge that predictiveness is distributed across many features. There is no clear threshold that marks these as significantly more important than others. The more even distribution of importance values implies that the model's predictive power is spread across many features, not just the top few. This is worth acknowledging as it speaks to the complexity of the genetic basis for ROS resistance. Focusing on the top 2-3 features does not diminish the importance of the findings but suggests that the focus on reductases and cell wall-related genes is part of a broader landscape of predictive features. I would recommend the authors provide more justification for focusing on these top two features or emphasize the broader contribution of the remaining features.

6. Interpretation of SHAP Values:

The use of SHAP values to interpret local feature importance is a strength of the study, as it allows for species-specific predictions. However, more detail on how SHAP values influenced experimental validation choices would be beneficial. For instance, why were the mannosyltransferase and OYE gene families prioritized for validation over other features? Providing a more detailed rationale for selecting certain gene families for validation based on SHAP values would enhance the reader's understanding of the experimental approach.

7. Feature Redundancy in Cell Wall Genes:

The study highlights cell wall-related genes as key features in predicting ROS resistance, which is interesting but raises the question of functional redundancy. Many cell wall-related genes are known to have overlapping functions. How did the authors account for potential redundancy in gene families when interpreting their results? Addressing this could help clarify whether the identified features have unique contributions to ROS resistance or represent redundant pathways.

8. Temperature Sensitivity in ROS Stress (L278-280):

The observation that ROS stress sensitivity was higher at 30°C than at room temperature is intriguing and warrants further exploration (Fig. S3). While the authors note this as an "interesting" finding, providing more context would strengthen the discussion. Recent studies have demonstrated that intracellular ROS can be a primary growth inhibitor and even lethal to *Saccharomyces cerevisiae* under extreme temperature conditions, both at near-freezing temperatures (Laman Trip et al., *Nature Communications*, 2022) and high temperatures above 37°C (Laman Trip & Youk, *Nature Microbiology*, 2020). These studies showed that deleting key antioxidant genes, such as those encoding glutaredoxins and catalases, caused the most exacerbated growth inhibition at low temperatures. Moreover, glutathione was found to be necessary and sufficient for enabling yeast growth at high temperatures, further emphasizing the role of ROS in temperature-induced stress. Given these findings, the known effects of ROS at elevated and low temperatures could provide a valuable framework for interpreting the current study's findings on temperature sensitivity at 30°C. Moreover, the Introduction or Discussion sections could benefit from discussing how ROS impacts *S. cerevisiae* and other yeast species across different temperature ranges, thereby linking the temperature-dependent ROS stress to a broader context in yeast biology.

(Remarks on code availability)

Reviewer #4

(Remarks to the Author)

The biggest strength of this study is the novel dataset collected on various yeast species and their growth responses to different concentrations of the ROS-inducing agent TBOOH; making this dataset publicly available would enhance its value. The innovative methodology—linking gene family size with ROS resistance and training the model on gene counts per family—offers a novel approach for pinpointing specific enzymes related to ROS resistance. The experimental validation using non-conventional yeast, *Kluyveromyces lactis*, further demonstrates the method's cross-species applicability, adding depth to the study's findings.

General Feedback

Using only TBOOH as the ROS-inducing agent could bias results toward lipid-associated pathways; testing additional ROS-generating compounds (e.g., hydrogen peroxide, diamide) would provide a more comprehensive view of ROS resistance mechanisms.

For the MNT experimental validation, deleting only two paralogous mannosyltransferase genes does not adequately demonstrate the correlation between MNT gene count and ROS resistance; instead, selecting yeast species with approximately 10 MNT genes and progressively deleting them (e.g., reducing from 10 to 9, 8, 7 genes) and measuring growth would provide stronger evidence of this relationship.

Measuring the expression levels of OYE and MNT genes following upregulation or deletion would help confirm the correlation between gene family size and ROS resistance. This validation step would ensure that observed effects are directly attributable to these gene families.

Minor comments on introduction:

"There are still relatively few studies that have used ML in a macroevolutionary context, and these typically use a specific set of enzymes or curated annotations as features,"

The term "macroevolutionary context" may be unclear to some readers. A brief definition or context within the introduction would enhance clarity.

Major comments on results:

Section: Identification of ROS-resistant and ROS-sensitive species

Could you clarify the rationale for choosing TBOOH over other ROS inducers like diamide, menadione, or hydrogen peroxide? Given that TBOOH specifically targets lipids and membranes, this may introduce a bias toward lipid-associated pathways, potentially explaining the prominent role of MNT genes, which are involved in lipid-linked biosynthesis. Testing additional ROS compounds may help establish whether the identified gene families (e.g., OYE and MNT) consistently emerge as central across different types of ROS stress.

<https://www.pnas.org/doi/10.1073/pnas.0305888101> This study examines how *Saccharomyces cerevisiae* deletion mutants react to various reactive oxygen species (ROS) to uncover key cellular components in oxidative stress resistance. The key

pathways for ROS resistance were mechanisms involving mitochondrial function, pentose phosphate pathway, ergosterol and lipid metabolism, vacuolar protein sorting.

It would be informative to determine if your model detects these pathways when analyzing GO terms, as these pathways are essential in ROS resistance. If key pathways are absent, please consider adding them to assess their influence on SHAP values and predictive accuracy.

The paper demonstrates that different ROS types, such as hydrogen peroxide, superoxide, lipid peroxides, and diamide, uniquely impact cells, each requiring specific cellular defenses. For example, hydrogen peroxide tolerance depends largely on mitochondrial respiratory function, whereas resistance to diamide relies on vacuolar protein sorting and acidification, indicating that each ROS engages distinct protective pathways.

This is why I'm proposing using a different ROS generator to conduct a similar EAUC profile. Would the SHAP values and gene family profiles remain consistent, and would key components like OYE and mannosyl transferases still emerge as central if hydrogen peroxide were used?

Section: The gene family encoding the reductase old yellow enzyme contributes to ROS resistance

Figure 5A does not clearly differentiate OYE gene counts between ROS-resistant and ROS-sensitive species. A single comparison compiling ROS-sensitive and ROS-resistant species with corresponding OYE gene counts, including statistical analysis, would strengthen the evidence. Further investigation into whether similar patterns apply to other reductases, such as OYE2 and OYE3, could also clarify whether the observed trends in OYE gene counts are consistent across related enzymes.

Section: The gene family encoding the reductase old yellow enzyme contributes to ROS resistance

It would be beneficial to experimentally verify whether introducing multiple copies of OYE genes enhances ROS resistance more effectively than simply upregulating the OYE gene under a stronger promoter. This would strengthen the model's prediction that increasing gene copy number is directly linked to ROS resistance.

Section: A novel role for mannosyl transferases in yeast ROS resistance

Similar idea for Figure 5a, do the same thing with Figure 6a.

Deleting only two paralogous mannosyltransferase genes is insufficient to demonstrate a correlation between MNT gene number and ROS resistance. Instead, consider selecting three yeast species with approximately 10 MNT genes (as shown in Figure 6B) and progressively deleting them (e.g., reducing from 10 to 9, 8, 7 genes) to assess the impact on growth under ROS stress.

Minor comments on results:

Section: Identification of ROS-resistant and ROS-sensitive species

In Figure 1a, marking the 20% poorest-growing and 20% best-growing species with a symbol (e.g., an asterisk) in Figure 1A would improve clarity, helping readers quickly identify ROS-sensitive and ROS-resistant groups.

Section: Understanding the local influence of gene families on the classification model

"local influences were estimated using SHAP values (Table S3), where positive and negative values indicate features contributing to ROS-sensitive and resistant predictions, respectively." This should be the other way around. Positive SHAP = ros resistant and negative SHAP = ros sensitive

Section: Functional characteristics of predictive gene families

Define how you calculated fold enrichment as shown in Figure 3 in your Methodology

Section: The gene family encoding the reductase old yellow enzyme contributes to ROS resistance

"K. lactis had a negative SHAP value for OYE, which indicates that this gene family pushed K. lactis towards a ROS-sensitive classification." Please clarify why OYE, typically associated with increased ROS resistance, would result in a negative SHAP value.

General major comments:

When upregulating OYE or down regulating MNT genes, measuring the resulting enzyme levels could validate that the effects observed are specifically attributable to these gene families. Confirming these levels would strengthen the connection between gene family size and ROS resistance.

Similar to the detailed analysis conducted for OYE, it would be informative to explore SHAP values for specific mannosyltransferases within the MNT gene family. This analysis could help pinpoint individual enzymes that contribute most significantly to ROS resistance.

Additional comments:

Supplemental Tables not provided

(Remarks on code availability)

Version 1:

Reviewer comments:

Reviewer #1

(Remarks to the Author)

The authors have done a commendable job in my opinion of addressing my concerns and those of the other reviewers. I believe the authors have proved the utility of their ML model using t-BOOH as one example of oxidative stress to make cross-species predictions. This on its own is an important proof of concept, and any relevant stress could have been used to make this important point (osmotic, heat, etc.). I think the authors have improved the manuscript by clarifying their choice of t-BOOH as a stressor and using more specific language about t-BOOH resistance vs general oxidative stress resistance where appropriate, with added experiments using hydrogen peroxide and menadione a good faith effort to help clarify that issue. I also appreciate the use of proteomics to validate overexpression and deletion of candidate genes, which strengthened the evidence for those interpretations.

(Remarks on code availability)

Reviewer #2

(Remarks to the Author)

I am satisfied with the explanations on oxidative stress and convinced by the additional figures added. This manuscript is ready to be published as it is.

(Remarks on code availability)

Reviewer #3

(Remarks to the Author)

The authors have addressed all my major comments, and in my assessment, they have also sufficiently addressed the key concerns raised by the other reviewers.

The revised manuscript strengthens the rigor and depth of an already outstanding study, both in terms of novelty and technical execution.

I recommend publication without further revision and congratulate the authors on their excellent work.

(Remarks on code availability)

Informative readme file, including appropriate shell commands to run the files in conda environment. A very minor suggestion: include which version of python (3.12?) was used for running your codes.

Reviewer #4

(Remarks to the Author)

Authors addressed my comments, I don't have anymore comments.

(Remarks on code availability)

It is very hard to test this, journals should do this job themselves.

REVIEWER COMMENTS and responses

Reviewer #1 (Remarks to the Author):

In this manuscript, Aranguiz et al. use a Random Forest ML algorithm to identify gene families that predict high ROS resistance across diverse yeast species spanning a large phylogenetic space. The authors measured levels of ROS resistance via plate reader growth analysis of 285 different yeast species on two different concentrations of tert-butyl hydroperoxide, and then used the 20th percentile tails of the distribution to define species as ROS-sensitive or resistant. They applied a ML algorithm that used the phenotype data and previously generated orthogroup assignments across the species to identify gene families whose size correlated with ROS resistance. Two of the top families included reductases and genes involved in cell wall synthesis, and the authors validated the predictive value of the model by over-expression of an old yellow enzyme reductase in an ROS-sensitive species, as well as deletion of mannosyltransferase-encoding genes in ROS-resistant *Saccharomyces cerevisiae*.

I think this paper very nicely illustrates how ML models can leverage comparative genomics and phenotype data to make predictions about genotype-phenotype relationships. I think this paper is of broad interest, in addition to being well written with a clear and concise logic throughout. I do not have any major issues, and have only a few minor comments.

We thank the reviewer for their appreciation and supportive comments about manuscript.

I should point out that my expertise is in fungal genetics/genomics and stress biology, and not ML algorithms. As such, I could conceivably miss any issues with the ML model. With that caveat in mind, the AUC-ROC and F1 scores both point to a model that has reasonably strong predictive quality for the testing dataset. Based on the held out test data, it appears that the model has an ~10% false positive rate and an ~15% false negative rate, which from my perspective seems pretty good. I do wonder how certain choices may influence model performance, such as classifying ROS-sensitive and resistant species based on the 20th percentile tails instead of the 10th percentile (or 30th), or if phenotype data was entered as relative EAUC values instead of classifications. Another minor caveat of the model in my mind is that the authors leveraged genes that were known to be upregulated by oxidative stress when choosing candidates for over-expression and knock-out experiments. That's a sound strategy that turned out to be fruitful, but it seems like that could be included in the model to either rank genes within a family, or to even be used on the front end to help rank the families beyond family size. I bring these things up not necessarily for changing the model for this particular paper, but because I hope the authors might consider further iterations down the road. This seems like something that could be a useful standalone software

package for the fungal community, and I also hope that the authors consider releasing the model in more accessible format beyond what is on the github repository in the future. I can imagine a package that allows users to input phenotype data and then outputs top features and their SHAP values would be widely used. Please take this overall comment as 'food for future thought' and not something that I expect be added to this paper.

We appreciate these thoughtful comments and are certainly interested in developing some of these ideas in future studies. In particular, we have tried some regression analyses, rather than classification models, without much success. We may attempt to build regression models in future studies with more data. We have also considered how to include additional features, such as transcriptional data, but since data are only available for limited species, the model would likely be more *S. cerevisiae*-centric than is desirable. Future studies may attempt to incorporate this additional information and these approaches.

This model was largely based on the ML-pipeline developed by Shin-Han Shiu's lab <https://github.com/ShiuLab/ML-Pipeline>. A workshop is available on this GitHub page, which can be followed and implemented by anyone with some knowledge of python. To better publicize this resource, in addition to the manuscript-specific GitHub link, we have added the following to the Code Availability statement:

“A more general workshop and pipeline are also available on GitHub (<https://github.com/ShiuLab/ML-Pipeline>).”

Minor comments:

1) Line 141-142: I think it's important when thinking about variation in ROS resistance among pathogens that their in vivo ROS resistance may be different from their ROS resistance under laboratory conditions, as they may transcriptionally induce different ROS defenses in vivo.

This point is an important one that we have also considered. There are substantial technical limitations to performing such a high-throughput screen in vivo, but we hope that this study spurs more detailed work on various growth conditions by the fungal pathogen community. We have made the following caveat to Line 148:

“Although these data are intriguing, we note that these pathogens may respond to ROS differently in vivo than in our in vitro condition.”

2) Were the strains used for assaying ROS resistance the exact same genotype as the sequenced isolate? I mention this because gene number can vary across strains within

a species, and variation in the number of ENA genes in *S. cerevisiae* is notably linked to variation in salt resistance across strains.

For most species used in building the model in this study (101/114) we used the exact same genotype, the taxonomic type strains, for both phenotyping and their genomes for the model. The remaining species were generally those for which there was a higher quality genome publicly available. We have explicitly added this information to Table S1 under the column "Type_strain_match." We also agree that evaluating strain heterogeneity in ROS resistance within a single species would be very interesting and potentially quite powerful, particularly in the context of the pathogenic yeasts. However, that dramatic scope expansion would necessitate one or more separate projects and manuscripts.

3) It would be useful to compare family predictions to genes with known ROS sensitivity from deletion library screens in *S. cerevisiae* (e.g., DOI [10.1016/j.freeradbiomed.2007.12.008](https://doi.org/10.1016/j.freeradbiomed.2007.12.008)). The authors have a great example of redundancy masking the effects of KTR2 and YUR1 single deletions, but it would be interesting to know if there are families that are likely systematically underrepresented in deletion library screens.

This point is interesting to consider, although it is hard to compare systematically across all genes and species. We compared the z-scores from deletion library screens in *S. cerevisiae* grown in hydrogen peroxide for our top genes of importance ([DOI: 10.1126/sciadv.adg570](https://doi.org/10.1126/sciadv.adg570)) to the entire set of deletions and, in general, our list of important genes was not enriched for negative z-scores (sensitivity to hydrogen peroxide). In fact, the mean z-score was higher for our list of important genes, which suggests that their single deletions are less sensitive to hydrogen peroxide. Although these data are based on growth in H₂O₂ and not TBOOH, these results further highlight how the importance of these genes in large gene families may have been missed in single-gene deletion studies.

Density plot of the z-score of single gene knockouts of the important genes used in our ML model (red) and the remaining genes in the knockout collection (blue). Single knockouts of our top genes of importance are not more sensitive to H₂O₂ than the remaining genes. We have provided this figure for the interested reviewer, but we felt including it in the manuscript would distract from more central points.

4) Discussion: paragraph starting at 385. I do not think it is entirely fair to say that cell-wall related genes have not been previously identified as important for yeast ROS resistance. There are a number of papers that have found connections between the CWI pathway and oxidative stress (DOIs 10.1128/EC.00245-10, 10.1074/jbc.M411062200, 10.1016/j.fgb.2018.03.002), as well as potentially direct connections between cell wall protecting against oxidative stress resistance (DOIS 10.1128/ec.2.5.1018-1024.2003, 10.1016/j.febslet.2004.10.090, 10.1074/jbc.M411062200). There seem to also be connections to mannosylation and oxidative stress resistance (DOIs 10.1016/j.fgb.2012.08.005, 10.1016/j.bbrc.2006.12.012). It is fair to say that the involvement of KTR2 and YUR1 is a novel validation of the prediction, but I think it is somewhat overstating the case that cell wall and mannosylation involvement in oxidative stress resistance is a novel connection.

As noted, there is previous work connecting the cell wall to oxidative stress resistance. Most of the research done on oxidative stress and the cell wall connects the activation of the cell wall integrity pathway to oxidative stress. However, direct mechanisms are not especially well described. An exception to this knowledge gap is the important study showing that protein O-mannosyltransferases were required for oxidative stress responses through mannosylation of the sensor Mtl1, thereby showing a direct connection, and we thank the reviewer for bringing this to our attention. We have added additional references to the connections between the cell wall integrity pathway and specifically highlighted previous evidence about mannosyltransferases in the response

to oxidative stress. Beginning on Line 433 of the Discussion, we have added additional discussion of this literature:

“It has been established that there is cross-talk between the response to oxidative stress and the cell wall integrity pathway^{65–69}. Cell wall changes associated with increased permeability to H₂O₂ are also likely to increase susceptibility to oxidative stress⁷⁰. However, few studies in yeast have directly shown that mutants lacking cell wall biosynthetic enzymes have an impact on survival in oxidative stress. Interestingly, the protein O-mannosyltransferases Pmt1 and Pmt2 were found to support growth in TBOOH through their roles mannosylating a sensor of oxidative stress, Mtl1⁷¹. Outside of the yeasts, mannosyltransferases have been found to support ROS resistance in the entomopathogenic fungus in the subphylum Pezizomycotina, *Beauveria bassiana*⁷². We suspect that, in yeasts, the importance of large gene families contributing to ROS resistance, such as those encoding N-mannosyltransferases, has been overlooked due to redundancy, including in genetic model systems.”

We have also modified the header in the relevant Results section to specify that the novel role we identified is specific to N-mannosyltransferases.

Reviewer #1 (Remarks on code availability):

I did not run the code, but I did look at the code and documentation, and there appears to be enough information for someone moderately versed in python to run the code.

We thank the reviewer for confirming code accessibility.

Reviewer #2 (Remarks to the Author):

“Machine learning reveals genes impacting oxidative stress resistance across yeasts” by Aranguiz et al. describes tert-butyl hydroperoxide tolerance across 285 species of yeasts. This is an impressive phenotypic analysis across all the phylogenetic tree of single cell fungi. Using machine learning the authors were able to pinpoint the role of gene families such Old Yellow Enzyme and mannosyltransferases in such phenotype. The experimental validation of the expected families indicates that this approach is valid to identify new players in any phenotype with potential biotechnological or clinical implications, particularly those involved families of redundant genes that may be ignored in systematic deletion analysis.

We thank the reviewer for appreciating the breadth, insight, and validation in the manuscript.

However, the choice of t-BOOH as oxidative stress inductor is not obvious and it has to be explained regarding molecular mechanisms or experimental conditions. Hydrogen

peroxide would be the most obvious choice. It is indicated three times in the text talking about oxidative stress induced genes and tolerance mechanisms (308, 380 and 400). This is a molecule all cells produce naturally, and it is known in *S. cerevisiae* that different peroxiredoxins deal with H₂O₂ and alkyl peroxides, and the signature of the stress response is different (for instance see Iwai K, Naganuma A, Kuge S, 2010. Peroxiredoxin Ahp1 acts as a receptor for alkylhydroperoxides to induce disulfide bond formation in the Cad1 transcription factor. *J. Biol. Chem.* 285: 10597–10604). A superoxide generator like menadione would mimic better the burst in superoxide anions generated by macrophages to deal with pathogenic yeast. The genetic modifications described in Figures 5 and 6 that were tested for tert-butyl hydroperoxide tolerance have to prove their role in hydrogen peroxide and other stressors to prove that the ROS resistance that was indicated in the abstract (where t-BOOH was not indicated) is real. Besides, the impact of *S. cerevisiae* mannosyltransferases deletion is clear, but the mutant was not “hypersensitive” as indicated in the Abstract (line 40) or had “strong impact on ROS resistance” as mentioned in line 392.

We appreciate the question about why we chose TBOOH as our oxidizing agent. The main rationale was the technical consideration that TBOOH is a more stable compound than H₂O₂ (Koubek et al., 1963, <https://doi.org/10.1021/ja00898a016>), and different yeast species show high variation in lag phase in our control condition (up to 5 days). Therefore, we did not want the yeasts to be exposed to different effective concentrations of the oxidizing agent during active growth, especially since hydrogen peroxide decomposes more rapidly when glucose is present (Schubert and Wilmer 1991, [https://doi.org/10.1016/0891-5849\(91\)90135-P](https://doi.org/10.1016/0891-5849(91)90135-P)).

We have made several adjustments to clarify that we are talking about TBOOH sensitivity in our Abstract on Line 43:

“Here, we characterized the variation in resistance to the ROS-inducing compound tert-butyl hydroperoxide across the ancient yeast subphylum Saccharomycotina and used machine learning (ML) to identify gene families whose sizes were predictive of ROS resistance.”

We also clarified the scope in the Introduction on Line 116:

“In this study, we took a genome-wide comparative approach training an ML model to identify gene families predictive of high resistance to the ROS-inducing compound tert-butyl hydroperoxide across the yeast subphylum using all orthologous gene counts as features.”

We have also added a justification for why we used TBOOH in our Methods on Line 505:

“TBOOH was chosen as the oxidative stress agent since our yeast species have variable lag phases, and H₂O₂ decomposes rapidly, particularly in the presence of sugars^{79,80}. If we had chosen H₂O₂ for the screen, some slow growing yeasts would not have been exposed to the same effective concentration of oxidizing agent during their exponential growth phase.”

To further address the generality of our findings, we conducted additional validation experiments with other oxidative stressors using the modified strains overexpressing old yellow enzyme and deleting mannosyltransferase-encoding genes. Specifically, we found that the overexpression of old yellow enzyme improved growth in the presence of both hydrogen peroxide and menadione, suggesting that this enzyme is likely broadly important for ROS resistance. The mannosyltransferase results were more nuanced. The single knockout of the mannosyltransferase *KTR2* grew significantly worse than the wild type in hydrogen peroxide, whereas the *YUR1* deletion strain and double knockouts were not significantly different. Furthermore, these mutants did not grow significantly differently from wild type in the presence of menadione. These new results highlight that the impact of the genes we have identified differs, in some cases, depending on the source of ROS. These data have been added in Figures S5 and S8, and we added on Line 299:

“To assess the general importance of old yellow enzyme in ROS resistance, we also grew these strains in the presence of two other ROS generators, hydrogen peroxide and menadione. Consistently, we found that additional episomal copies of *KYE1* significantly improved the growth of *K. lactis* compared to the *GFP*-expressing control in both H₂O₂ and menadione (Fig. S5).”

To incorporate the mannosyltransferase results into the text, we added on Line 358:

“We also tested the sensitivity of these mutants to hydrogen peroxide and menadione as additional sources of oxidative stress. Interestingly, only the deletion mutant lacking *KTR2* grew significantly worse than the wild type in the presence of hydrogen peroxide, whereas all strains grew similarly in the presence of menadione (Fig. S8). Overall, these results suggest that the importance of some mannosyltransferases is dependent on the source of oxidative stress. However, *KTR2* may be generally important for yeast growth in the presence of multiple peroxides.”

We also added to the Discussion on Line 456:

“Through our comparative approach, we identified gene families encoding cell wall-modifying enzymes that are associated with ROS resistance and confirmed the importance of mannosyltransferases in resistance to TBOOH. However, future work is required to determine the exact mechanisms by which mannosyltransferases contribute

to resistance against TBOOH and why the source of ROS stress impacts the importance of mannosyltransferases.”

We have also described these new experiments in the Methods section beginning on Line 639:

“Impact of additional oxidative stress agents

For the *K. lactis* experiments investigating the old yellow enzyme, strains were grown in SC medium made with 1% monosodium glutamate as the nitrogen source supplemented with 200 µg/ml G418 sulfate at 30°C overnight in biological quadruplicate. Harvested cells were washed once with sterile water and normalized to OD_{600nm} = 1. Liquid growth assays were conducted in 96-well plates in 200 µl SC+MSG+G418 (supplemented with 2 mM H₂O₂ or 30 µM menadione) inoculated at OD_{600nm} = 0.1. Plates were placed into the SPECTROstar Omega Plate Reader with a Microplate Stacker and read every hour at OD_{600nm} for 72 hours.

For the *S. cerevisiae* experiments investigating mannosyltransferases, strains were grown in SC medium at 30°C overnight in biological quadruplicate. Harvested cells were washed once with sterile water and normalized to OD_{600nm} = 1. Liquid growth assays were conducted in 96-well plates in 200 µl SC (supplemented with 1 mM H₂O₂ or 30 µM menadione) inoculated at OD_{600nm} = 0.1. Plates were placed into the SPECTROstar Omega Plate Reader with a Microplate Stacker and read every hour at OD_{600nm} for 48 hours.”

Minor points

Explain what “FAD transport” in line 195 is referred to.

Thank you for noticing this strange editing error, which should have been “metal transport” and refers to the enrichment of copper and transition metal transport functions.

OYE enzymes have been involved in the reduction of citral to citronellal, and event of biotechnological relevance. Are the OYE genes also included in the category “aldehyde reductases of Figure 3A.

No, the OYE is the second gene family (OG0000030), while the first gene family (OG0000006) includes other aldehyde reductases (i.e. YDR541C, YGL039W, YGL157W, YOL151). For each gene family in Fig 3A, the respective genes in *S. cerevisiae* and *C. albicans* are listed in Table S2.

Reviewer #3 (Remarks to the Author):

1. Summary:

This study explores the use of gene family size as a feature for machine learning

models to predict phenotypic traits in yeast species, specifically their sensitivity or resistance to reactive oxygen species (ROS). The authors developed a Random Forest (RF) classifier using gene family size to predict ROS resistance across over 100 yeast species. The model highlighted reductase and cell wall-related gene families as the most predictive features. The study's key biological finding is that gene families encoding reductases were among the most predictive features, and the top 50 features were enriched for cell wall-related gene families. Experimental validation confirmed that increasing the copy number of genes encoding old yellow enzyme (OYE) increased ROS resistance in *Kluyveromyces lactis*, while deleting mannosyltransferase genes in *Saccharomyces cerevisiae* decreased ROS resistance. Overall, this study demonstrates a novel approach to connecting gene family size with trait variability in yeast, with potential applications in both biotechnology and antifungal development.

2. Key strengths

The manuscript is well-written, and the experimental work is thorough, especially the ROS resistance assays involving over 100 yeast species. The integration of machine learning with comparative genomics is a powerful approach that provides new insights into the genetic basis of ROS resistance. The figures, despite being dense and with fonts that will be too small to read in print (but okay when zoomed in), are still visually clear and convey the experimental scope and effort behind the work.

Although the machine-learning algorithms used in this study is not new - random forest with SHAP is one of the most frequently used Python packages - the novelty of using gene family size as an input feature for a machine learning model is compelling, and the study demonstrates the utility of this approach for identifying key genetic components related to stress resistance in yeast. The findings have practical applications in biotechnology (e.g., improving yeast strains for industrial production) and clinical settings (e.g., targeting pathogenic yeasts). The validation experiments, particularly the overexpression and deletion studies, solidify the model's predictions and provide valuable biological insights.

We thank the reviewer for the appreciation of the study's novelty, rigor, and breadth.

3. Overall recommendation:

I recommend publication with minor revisions. But I do have some recommendations for the authors. The authors can decide whether to address these if they don't push the study beyond its current scope. I don't believe any of these points require additional experiments but might require some rewriting and further analyses of their existing data.

We thank the reviewer for recognizing the significant advances and effort of the present work. Most comments could indeed be addressed with minor text revisions, but we did perform some additional validation experiments in response to Reviewers #2 and #4.

We invite the reviewer to read those responses if they are interested in these new data.

4. Major points:

1. L167-168: RF Feature Selection Algorithm Clarification:

The description of the “RF feature selection algorithm” is somewhat unclear. It would benefit from additional details about the feature selection process. Did the authors rely on SHAP values from the RF model to rank the features? Also, while gene family size is a primary input feature, how many total features were used in the model, and what was the architecture of the RF classifier (e.g., the number of trees, the depth of trees)? A better explanation of the complexity and interpretability of the RF model would help readers understand how the gene family size relates to the model's predictions. Adding this information would clarify the modeling process and emphasize the significance of the selected features.

We thank the reviewer for the advice to provide more details about the RF feature selection. We have updated the Results section to specifically state that Gini importance was used to select the most important features on Line 175:

“An RF feature selection algorithm was used to identify the 50 most important features based on Gini importance. Using these 50 features, an RF classifier was trained with 90% of species, and 10% were held out for testing.”

The pipeline we used (<https://github.com/ShiuLab/ML-Pipeline>) includes automated hyperparameter tuning to determine the best architecture for classification. However, to improve transparency, we have updated the Methods section to include information on the number of trees and their depth selected through this process on Line 535:

“To choose the most predictive features from the orthogroup matrix, we performed feature selection using the Random Forest (RF) algorithm specifying the number of features to select as 50 and the type as classification. The 50 most important features were selected based on Gini importance, which represents the normalized contribution of each feature to the purity of child nodes across all trees in each model. Using these 50 features, we ran an RF classification algorithm using a full grid search to determine the best combination of the number of estimators to use, the number of features to consider when looking for the best split, and maximum tree depth. A total of 500 trees with a maximum depth of 10 were used for each model. For each of 100 model replicates, a 10-fold cross-validation scheme was applied to assess the model's generalization ability.”

2. Clarity of RF Model Setup (related to above):

The manuscript could benefit from a clearer explanation of how gene family size was integrated into the model and how feature selection was performed. Was there any pre-

filtering of gene families before they were included as features, or were all gene families across all species considered in the model?

We did not use any pre-filtering strategies. All gene families were considered by the feature selection algorithm. We have updated the Methods on Line 525 to clarify:

“This class column was combined with the orthogroup matrix containing counts for all gene families across all species (Table S4), which was previously generated for the subphylum Saccharomycotina using OrthoFinder.”

3. Use of Random Forest (RF) model:

The application of RF classifier in this study was well-suited for identifying gene families predictive of ROS resistance. However, the authors could have explored additional machine learning models to compare performance, such as gradient-boosting machines or support vector machines. RF is a solid choice and it is one that is perhaps more friendly and easier to use for those new to machine learning, making this work broadly appealing to readers. But a broader exploration of other machine-learning models might reveal further insights. The authors might consider justifying why they chose RF instead of other ML models. If they explored other ML models, explain how they fared, in terms of predictability and interpretability, compared to the RF model.

We did consider other machine learning model algorithms as described in the workshop and recommendations from <https://github.com/ShiuLab/ML-Pipeline>, including primarily SVM and Fisher’s exact tests. We also tried different combinations of algorithms for feature selection and classification. From some preliminary tests, the random forest model seemed to have the most promising performance metrics, so we focused on the RF algorithm for further optimization. As noted by the reviewer, the random forest model also appealed to us as it has a good balance between robustness and interpretability, particularly when coupled with SHAP analyses to understand local influences of features. Ultimately, for this manuscript, we decided to focus on the novel biology revealed by the model, rather than performing a detailed comparison of different models.

4. Fig. 2C: Include Recall Value:

The manuscript provides metrics such as the F1 score and precision for the RF classifier but omits the recall value. Including the recall value would give a more complete picture of the model's performance, especially in understanding how well it identifies true positives. Additionally, showing the confusion matrix would help readers grasp the balance between false positives and false negatives in the classification model.

We have updated Fig 2c to include the recall values for both the validation (0.87) and the test set (0.86). Panel B of this figure also includes the mean balanced confusion

matrix, which was computed by averaging the confusion matrices from 100 iterations using unique sets of balanced instances. This approach helped us to understand the balance between false positives and false negatives across multiple replicates. We have added a description to the methods to clarify what this matrix represents on Line 545:

“Model performance was evaluated using the tradeoff between precision and recall (F1) statistics and the area under the curve of both the receiver operating characteristic (AUC-ROC) and the precision-recall (AUC-PR). The mean balanced confusion matrix for the validation set was calculated by averaging the confusion matrices from 100 independent model training runs, and the model was trained on unique subsets of the data for each iteration.”

5. Fig. 3A: Justification for Focus on Top Features:

The feature importance plot in Fig. 3A shows that the top two features (gene families encoding reductases) have an importance value of around 0.06, while many other features in the top 50 have values around 0.02—a difference of only about 3-fold. This suggests that many features contribute to the model’s performance. While the authors focus on the top two features, it would be fairer to acknowledge that predictiveness is distributed across many features. There is no clear threshold that marks these as significantly more important than others. The more even distribution of importance values implies that the model’s predictive power is spread across many features, not just the top few. This is worth acknowledging as it speaks to the complexity of the genetic basis for ROS resistance. Focusing on the top 2-3 features does not diminish the importance of the findings but suggests that the focus on reductases and cell wall-related genes is part of a broader landscape of predictive features. I would recommend the authors provide more justification for focusing on these top two features or emphasize the broader contribution of the remaining features.

We appreciate this comment, which highlights an important point. Our model suggests that ROS resistance is not mainly determined by a single gene family but rather has contributions from many different types of gene families. To provide a clearer rationale for our focus, we have revised our explanation for why we conducted a GO term analysis on the top 50 features in the Results on Line 199:

“Although these genes have functions that can be easily related to ROS mitigation, we noted that there was not a sharp decrease in importance after these two features, but rather a gradual and subtle decrease (Fig. 3a). Therefore, many of the other top features are also contributing to the correct classifications, suggesting that ROS resistance is complex and polygenic. This result prompted us to further interrogate the potential functions of the top 50 gene families by performing a gene ontology (GO) term enrichment analysis using all of the genes in the top 50 orthogroups from *S. cerevisiae*

and *C. albicans* because these species have the most well-annotated genomes in our dataset.”

We have also highlighted the importance of many different gene families in the Discussion on Line 487:

“This work used a rich genomic dataset to identify previously overlooked gene families predictive of ROS resistance, such as the *N*-mannosyltransferase-encoding family, and it underscores the complexity of ROS resistance because several different gene families contribute to correct prediction of ROS resistance across yeast species.”

6. Interpretation of SHAP Values:

The use of SHAP values to interpret local feature importance is a strength of the study, as it allows for species-specific predictions. However, more detail on how SHAP values influenced experimental validation choices would be beneficial. For instance, why were the mannosyltransferase and OYE gene families prioritized for validation over other features? Providing a more detailed rationale for selecting certain gene families for validation based on SHAP values would enhance the reader’s understanding of the experimental approach.

Several technical considerations went into our validation choices. First, we needed to use species that were genetically tractable, and, ideally, had a reasonable number of genes to work with. We also looked at previous literature showing that these genes were either upregulated in response to ROS stress or were required for robust growth on oxidizing agents as cited in the manuscript (57, 63). However, to improve clarity, we have updated the Results section immediately preceding the validation experiments on Line 259:

“Altogether, SHAP values helped us interpret the local influence of features on ROS resistance class predictions. Since our most important gene families included many reductases and cell wall related enzymes, we used SHAP values to guide the selection of gene-species combinations for functional validation experiments of the *OYE* and *MNT* gene families. These validation experiments are ultimately vital for generating biological understanding and subsequent application of these results.”

7. Feature Redundancy in Cell Wall Genes:

The study highlights cell wall-related genes as key features in predicting ROS resistance, which is interesting but raises the question of functional redundancy. Many cell wall-related genes are known to have overlapping functions. How did the authors account for potential redundancy in gene families when interpreting their results? Addressing this could help clarify whether the identified features have unique contributions to ROS resistance or represent redundant pathways.

We expect that redundant features would be included in the top features because, in the random forest model, random subsets of features are used to create unique decision trees. This approach helps to mitigate the chance that we missed an important feature because of its redundancy. However, it is possible that some features we have identified, particularly the cell wall-related gene families, may not be functionally important for ROS resistance, but rather highly correlated with other cell wall-related gene families required for ROS resistance. Although the gene families we identified should be considered candidates for their importance in ROS resistance, functional validation is key, in part because of this potential redundancy. We hope that, through the publication of these gene lists, other groups will see these data and be motivated to test additional genes in their favorite organisms.

8. Temperature Sensitivity in ROS Stress (L278-280):

The observation that ROS stress sensitivity was higher at 30°C than at room temperature is intriguing and warrants further exploration (Fig. S3). While the authors note this as an "interesting" finding, providing more context would strengthen the discussion. Recent studies have demonstrated that intracellular ROS can be a primary growth inhibitor and even lethal to *Saccharomyces cerevisiae* under extreme temperature conditions, both at near-freezing temperatures (Laman Trip et al., Nature Communications, 2022) and high temperatures above 37°C (Laman Trip & Youk, Nature Microbiology, 2020). These studies showed that deleting key antioxidant genes, such as those encoding glutaredoxins and catalases, caused the most exacerbated growth inhibition at low temperatures. Moreover, glutathione was found to be necessary and sufficient for enabling yeast growth at high temperatures, further emphasizing the role of ROS in temperature-induced stress. Given these findings, the known effects of ROS at elevated and low temperatures could provide a valuable framework for interpreting the current study's findings on temperature sensitivity at 30°C. Moreover, the Introduction or Discussion sections could benefit from discussing how ROS impacts *S. cerevisiae* and other yeast species across different temperature ranges, thereby linking the temperature-dependent ROS stress to a broader context in yeast biology.

While we agree that temperature is certainly an interesting aspect of yeast ROS resistance and may have particular importance for the pathogenic yeasts, we are hesitant to discuss this result too deeply because our original screen was only conducted at room temperature. Nonetheless, to highlight the influence of temperature on ROS sensitivity and other aspects that would make for interesting future studies, we have added a small section to the Discussion on Line 462:

“Although we validated the roles of two gene families predicted to be important in ROS resistance, this model has generated many hypotheses about other gene families that may be contributing to ROS resistance. We have calculated SHAP values for these gene-species combinations (Table S3), which could be used to guide future validation experiments and applications in specific yeasts. Furthermore, our validation

experiments revealed that yeasts were more sensitive to TBOOH at 30°C than at room temperature. This result is consistent with other reports that the oxidative stress response is required for survival at elevated temperatures⁷⁵⁻⁷⁷ and highlights the importance of exploring the interaction between ROS and temperature sensitivity, particularly in pathogenic yeasts, which face both in the context of host infection.”

Reviewer #4 (Remarks to the Author):

The biggest strength of this study is the novel dataset collected on various yeast species and their growth responses to different concentrations of the ROS-inducing agent TBOOH; making this dataset publicly available would enhance its value. The innovative methodology—linking gene family size with ROS resistance and training the model on gene counts per family—offers a novel approach for pinpointing specific enzymes related to ROS resistance. The experimental validation using non-conventional yeast, *Kluyveromyces lactis*, further demonstrates the method's cross-species applicability, adding depth to the study's findings.

General Feedback

Using only TBOOH as the ROS-inducing agent could bias results toward lipid-associated pathways; testing additional ROS-generating compounds (e.g., hydrogen peroxide, diamide) would provide a more comprehensive view of ROS resistance mechanisms.

For the MNT experimental validation, deleting only two paralogous mannosyltransferase genes does not adequately demonstrate the correlation between MNT gene count and ROS resistance; instead, selecting yeast species with approximately 10 MNT genes and progressively deleting them (e.g., reducing from 10 to 9, 8, 7 genes) and measuring growth would provide stronger evidence of this relationship.

Measuring the expression levels of OYE and MNT genes following upregulation or deletion would help confirm the correlation between gene family size and ROS resistance. This validation step would ensure that observed effects are directly attributable to these gene families.

We thank the reviewer for appreciating the novelty and breadth of the dataset, as well as our functional validation of its findings. We will respond to each of the 3 “General Feedback” points when they are raised in more detail below.

Minor comments on introduction:

“There are still relatively few studies that have used ML in a macroevolutionary context, and these typically use a specific set of enzymes or curated annotations as features,” The term “macroevolutionary context” may be unclear to some readers. A brief definition or context within the introduction would enhance clarity.

We have added a brief explanation of macroevolutionary context in our study on Line 110:

“While the application of ML across strains or mutants of a single species is being employed widely^{42–45}, there are still relatively few studies that have used ML in a macroevolutionary context (i.e. assessing large-scale evolutionary patterns across millions of years, such as an entire subphylum), and these studies often typically use a specific set of enzymes or curated annotations as features^{3,46–49}.”

Major comments on results:

Section: Identification of ROS-resistant and ROS-sensitive species

Could you clarify the rationale for choosing TBOOH over other ROS inducers like diamide, menadione, or hydrogen peroxide? Given that TBOOH specifically targets lipids and membranes, this may introduce a bias toward lipid-associated pathways, potentially explaining the prominent role of MNT genes, which are involved in lipid-linked biosynthesis. Testing additional ROS compounds may help establish whether the identified gene families (e.g., OYE and MNT) consistently emerge as central across different types of ROS stress.

<https://www.pnas.org/doi/10.1073/pnas.0305888101> This study examines how *Saccharomyces cerevisiae* deletion mutants react to various reactive oxygen species (ROS) to uncover key cellular components in oxidative stress resistance. The key pathways for ROS resistance were mechanisms involving mitochondrial function, pentose phosphate pathway, ergosterol and lipid metabolism, vacuolar protein sorting. It would be informative to determine if your model detects these pathways when analyzing GO terms, as these pathways are essential in ROS resistance. If key pathways are absent, please consider adding them to assess their influence on SHAP values and predictive accuracy.

The paper demonstrates that different ROS types, such as hydrogen peroxide, superoxide, lipid peroxides, and diamide, uniquely impact cells, each requiring specific cellular defenses. For example, hydrogen peroxide tolerance depends largely on mitochondrial respiratory function, whereas resistance to diamide relies on vacuolar protein sorting and acidification, indicating that each ROS engages distinct protective pathways.

This is why I'm proposing using a different ROS generator to conduct a similar EAUC profile. Would the SHAP values and gene family profiles remain consistent, and would key components like OYE and mannosyl transferases still emerge as central if hydrogen peroxide were used?

We appreciate the question about why we chose TBOOH as our oxidizing agent, a point also raised by Reviewer #2. Before we reproduce that response below, we will compare our data with other previously identified ROS-related genes with new integrative analyses in the figure below. Specifically, we considered specific genes, including those highlighted by Thorpe et al. 2004, which we expected to be important to ROS resistance

before using a ML model. These genes encoded enzymes that included catalases, superoxide dismutases, and peroxidases. However, in general, these gene families did not have strong correlations with ROS resistance (see figure below: no significant correlation with catalases, superoxide dismutase, cytochrome c peroxidase, or peroxiredoxin). We suspect that these gene families did not have predictive power because these genes are highly conserved. These gene families were all included in the input data matrix used to build the classifier model as well, but they were not selected by the RF feature selection algorithm as being effective for classification.

Here we reproduce the response to Reviewer #2 about our motivations for choosing TBOOH for our initial screen, as well as new experimental data using additional ROS generators.

The main rationale was the technical consideration that TBOOH is a more stable compound than H_2O_2 (Koubek et al., 1963, <https://doi.org/10.1021/ja00898a016>), and different yeast species show high variation in lag phase in our control condition (up to 5 days). Therefore, we did not want the yeasts to be exposed to different effective concentrations of the oxidizing agent during active growth, especially since hydrogen peroxide decomposes more rapidly when glucose is present (Schubert and Wilmer 1991, [https://doi.org/10.1016/0891-5849\(91\)90135-P](https://doi.org/10.1016/0891-5849(91)90135-P)).

We have made several adjustments to clarify that we are talking about TBOOH sensitivity in our Abstract on Line 43:

“Here, we characterized the variation in resistance to the ROS-inducing compound tert-butyl hydroperoxide across the ancient yeast subphylum Saccharomycotina and used machine learning (ML) to identify gene families whose sizes were predictive of ROS resistance.”

We also clarified the scope in the Introduction on Line 116:

“In this study, we took a genome-wide comparative approach training an ML model to identify gene families predictive of high resistance to the ROS-inducing compound tert-butyl hydroperoxide across the yeast subphylum using all orthologous gene counts as features.”

We have also added a justification for why we used TBOOH in our Methods on Line 505:

“TBOOH was chosen as the oxidative stress agent since our yeast species have variable lag phases, and H₂O₂ decomposes rapidly, particularly in the presence of sugars^{79,80}. If we had chosen H₂O₂ for the screen, some slow growing yeasts would not have been exposed to the same effective concentration of oxidizing agent during their exponential growth phase.”

To further address the generality of our findings, we conducted additional validation experiments with other oxidative stressors using the modified strains overexpressing old yellow enzyme and deleting mannosyltransferase-encoding genes. Specifically, we found that the overexpression of old yellow enzyme improved growth in the presence of both hydrogen peroxide and menadione, suggesting that this enzyme is likely broadly important for ROS resistance. The mannosyltransferase results were more nuanced. The single knockout of the mannosyltransferase *KTR2* grew significantly worse than the wild type in hydrogen peroxide, whereas the *YUR1* deletion strain and double knockouts were not significantly different. Furthermore, these mutants did not grow significantly differently from wild type in the presence of menadione. These new results highlight that the impact of the genes we have identified differs, in some cases, depending on the source of ROS. These data have been added in Figures S5 and S8, and we added on Line 299:

“To assess the general importance of old yellow enzyme in ROS resistance, we also grew these strains in the presence of two other ROS generators, hydrogen peroxide and menadione. Consistently, we found that additional episomal copies of *KYE1* significantly improved the growth of *K. lactis* compared to the *GFP*-expressing control in both H₂O₂ and menadione (Fig. S5).”

To incorporate the mannosyltransferase results into the text, we added on Line 358:

“We also tested the sensitivity of these mutants to hydrogen peroxide and menadione as additional sources of oxidative stress. Interestingly, only the deletion mutant lacking *KTR2* grew significantly worse than the wild type in the presence of hydrogen peroxide, whereas all strains grew similarly in the presence of menadione (Fig. S8). Overall, these results suggest that the importance of some mannosyltransferases is dependent on the source of oxidative stress. However, *KTR2* may be generally important for yeast growth in the presence of multiple peroxides.”

We also added to the Discussion on Line 456:

“Through our comparative approach, we identified gene families encoding cell wall-modifying enzymes that are associated with ROS resistance and confirmed the importance of mannosyltransferases in resistance to TBOOH. However, future work is required to determine the exact mechanisms by which mannosyltransferases contribute to resistance against TBOOH and why the source of ROS stress impacts the importance of mannosyltransferases.”

We have also described these new experiments in the Methods section beginning on Line 639:

“Impact of additional oxidative stress agents

For the *K. lactis* experiments investigating the old yellow enzyme, strains were grown in SC medium made with 1% monosodium glutamate as the nitrogen source supplemented with 200 µg/ml G418 sulfate at 30°C overnight in biological quadruplicate. Harvested cells were washed once with sterile water and normalized to OD_{600nm} = 1. Liquid growth assays were conducted in 96-well plates in 200 µl SC+MSG+G418 (supplemented with 2 mM H₂O₂ or 30 µM menadione) inoculated at OD_{600nm} = 0.1. Plates were placed into the SPECTROstar Omega Plate Reader with a Microplate Stacker and read every hour at OD_{600nm} for 72 hours.

For the *S. cerevisiae* experiments investigating mannosyltransferases, strains were grown in SC medium at 30°C overnight in biological quadruplicate. Harvested cells were washed once with sterile water and normalized to OD_{600nm} = 1. Liquid growth assays were conducted in 96-well plates in 200 µl SC (supplemented with 1 mM H₂O₂ or 30 µM menadione) inoculated at OD_{600nm} = 0.1. Plates were placed into the SPECTROstar Omega Plate Reader with a Microplate Stacker and read every hour at OD_{600nm} for 48 hours.”

Section: The gene family encoding the reductase old yellow enzyme contributes to ROS resistance

Figure 5A does not clearly differentiate OYE gene counts between ROS-resistant and ROS-sensitive species. A single comparison compiling ROS-sensitive and ROS-resistant species with corresponding OYE gene counts, including statistical analysis, would strengthen the evidence. Further investigation into whether similar patterns apply to other reductases, such as OYE2 and OYE3, could also clarify whether the observed trends in OYE gene counts are consistent across related enzymes.

We thank the reviewer for this suggestion, which will make this figure easier to understand. We have added a boxplot in Figure 5 to directly compare the gene family sizes in ROS-resistant and ROS-sensitive species. The *S. cerevisiae* paralogs OYE2 and OYE3 are included in this gene family, which is specified in Supplemental Table 2 and discussed in the Methods on Line 586:

“*K. lactis* *KYE1* encodes OYE and is the species’ sole ortholog to the *S. cerevisiae* S288C reference genes *OYE2* and *OYE3*, as well as the original *OYE1* from hybrid lager yeast^{89,90}.”

Section: The gene family encoding the reductase old yellow enzyme contributes to ROS resistance

It would be beneficial to experimentally verify whether introducing multiple copies of OYE genes enhances ROS resistance more effectively than simply upregulating the OYE gene under a stronger promoter. This would strengthen the model's prediction that increasing gene copy number is directly linked to ROS resistance.

We apologize for any confusion about the design of this experiment. This experiment involved adding an additional copy of *KYE1* under a constitutive *TEF1* promoter. Thus, we added an extra episomal copy rather than upregulating the native copy. Fine-tuning the expression of *OYE* for optimal ROS resistance would be very interesting, and we are working on that experiment for a subsequent application-driven paper. We have added a sentence to Line 592 of the Methods to specify that our *K. lactis* background strain retained the native *KYE1* copy:

“Both expression plasmids, as well as an empty vector control, were transformed into *K. lactis*, which retained the native copy of *KYE1*, using the LiAc/heat shock method as previously described⁹³.”

Section: A novel role for mannosyl transferases in yeast ROS resistance

Similar idea for Figure 5a, do the same thing with Figure 6a.

Deleting only two paralogous mannosyltransferase genes is insufficient to demonstrate a correlation between MNT gene number and ROS resistance. Instead, consider selecting three yeast species with approximately 10 MNT genes (as shown in Figure 6B) and progressively deleting them (e.g., reducing from 10 to 9, 8, 7 genes) to assess the impact on growth under ROS stress.

Similar to our modification of Figure 5, we have added a boxplot to Figure 6 to directly compare the gene family sizes in ROS-resistant and ROS-sensitive species.

A detailed exploration of the whole gene family would be interesting. However, our goal here is to validate that the ML model identified gene families that are functionally important in resistance to TBOOH. The proposed experiment of picking a yeast with 10 *MNT* genes and sequentially deleting all *MNT* orthologs would be a serious undertaking, especially since many of these yeasts do not have established protocols for genetic modification. Furthermore, even systematically deleting members of an entire gene family only in *S. cerevisiae*, coupled with the analysis of the importance of its members, is an entire project in and of itself (e.g. Saint-Prix et al., 2004 <https://doi.org/10.1099/mic.0.26999-0>; Krause and Hittinger, 2022,

<https://doi.org/10.1093/molbev/msac202>). Therefore, we have changed our analysis in Figure 6 from a correlation to categorical comparisons between strains to make it clear that we do not have sufficient evidence to conclude that the number of deletions is correlated with a decrease in ROS resistance.

Minor comments on results:

Section: Identification of ROS-resistant and ROS-sensitive species

In Figure 1a, marking the 20% poorest-growing and 20% best-growing species with a symbol (e.g., an asterisk) in Figure 1A would improve clarity, helping readers quickly identify ROS-sensitive and ROS-resistant groups.

We thank the reviewer for this suggestion and have updated Figure 1 accordingly.

Section: Understanding the local influence of gene families on the classification model
“local influences were estimated using SHAP values (Table S3), where positive and negative values indicate features contributing to ROS-sensitive and resistant predictions, respectively. “ This should be the other way around. Positive SHAP = ros resistant and negative SHAP = ros sensitive

Thank you for noticing this error. We have corrected it on Line 239:

“The local influences were estimated by calculating SHAP values for the positive class (Table S3), where positive and negative values indicate features contributing to ROS-resistant and ROS-sensitive predictions, respectively.”

Section: Functional characteristics of predictive gene families

Define how you calculated fold enrichment as shown in Figure 3 in your Methodology

We have added a description of how enrichment was calculated to Line 564:

“Fold-enrichment was calculated by pantherdb as the number of genes in our list divided by the expected number from the background genes we provided.”

Section: The gene family encoding the reductase old yellow enzyme contributes to ROS resistance

“*K. lactis* had a negative SHAP value for OYE, which indicates that this gene family pushed *K. lactis* towards a ROS-sensitive classification.” Please clarify why OYE, typically associated with increased ROS resistance, would result in a negative SHAP value.

K. lactis has only one copy of the *OYE* gene, whereas many of our resistant yeasts have many *OYE* genes (mean =4.5, range =0-21, Figure S2). Therefore, it is the small

number of *OYE* genes that contributes to the negative SHAP value. We have modified our description to improve clarity on Line 285:

“Since this species only has one copy of the gene encoding old yellow enzyme, *KYE1*, we tested whether increasing the copy number of *KYE1* conferred ROS resistance. We noted that SHAP values increased with increasing *OYE* copy number (Fig. S2), so we expressed an additional episomal copy of *KYE1* under the control of the moderately strong *K. lactis* *TEF1* promoter to mimic the addition of multiple *KYE1* copies.”

General major comments:

When upregulating *OYE* or down regulating *MNT* genes, measuring the resulting enzyme levels could validate that the effects observed are specifically attributable to these gene families. Confirming these levels would strengthen the connection between gene family size and ROS resistance.

We performed new proteomic experiments to measure the abundance of proteins in whole cell lysate of our modified strains. In *K. lactis*, we validated that the levels of *Kye1* were increased in all three of our overexpressing strains (Fig. S3). In *S. cerevisiae*, the *Ktr2* protein had too low of abundance to identify, even in the wild-type strain, but we were able to confirm that *Yur1* was not detected in our deletion strains (Fig. S6).

Similar to the detailed analysis conducted for *OYE*, it would be informative to explore SHAP values for specific mannosyltransferases within the *MNT* gene family. This analysis could help pinpoint individual enzymes that contribute most significantly to ROS resistance.

We have calculated the SHAP values for all gene families in *S. cerevisiae* and found that the *MNT* gene family had a positive SHAP value (0.0352). Since our model was trained on gene family size, SHAP values cannot be calculated for individual genes within a family. We have added information about the SHAP value for *MNT* in the Results on Line 338:

“Therefore, we calculated SHAP values for *S. cerevisiae* (Table S4) and found that the *MNT* gene family had a positive SHAP value. This result suggests that the large size of the gene family was contributing to its positive classification, which led us to hypothesize that reducing the size of this family would impair ROS resistance, thus making the *MNT* gene family a promising target for validation experiments.”

Additional comments:

Supplemental Tables not provided

These tables were submitted to the journal, and we apologize that they were not

accessible to the reviewer. If there is a compatibility issue with the submission system, we would be happy to work with the editor to provide supplemental tables in an alternative format.